# Rare regulatory mutations disrupt mesenchymal molecular programs driving endocardial cushion formation in bicuspid aortic valve

Artemy Zhigulev [1,2] ✉, Andrey Buyan [3,4], Enikő Lázár [1], Nikita Gryzunov [3,5], Karin Lång[2], Raphaël Mauron [1], Vladimir Nozdrin [6], Rapolas Spalinskas[1], Sailendra Pradhananga[1], Madeleine Petersson Sjögren[1], Doreen Schwochow[1], Anders Franco-Cereceda [7], Joakim Lundeberg [1], Ivan V. Kulakovskiy [3,4,8], Per Eriksson[2], Hanna M. Björck [2] ✉ & Pelin Sahlén [1] ✉

Bicuspid aortic valve, a prevalent congenital malformation, predisposes individuals to severe complications. Although the condition exhibits substantial heritability, known protein-coding and common regulatory mutations explain a minority of cases. To assess the contribution of rare regulatory variants, here we integrate high-resolution three-dimensional genome organization profiling with matched whole-genome sequencing from eight individuals with bicuspid aortic valves and eight with standard tricuspid aortic valves. In bicuspid aortic valve patients, mutation-driven chromatin rewiring affected 1.8-fold more valve development genes than in healthy individuals. Genome-wide in silico analyses show that rare regulatory mutations disrupt the transcriptomes of mesenchymal cell populations necessary for endocardial cushion formation. We identify 198 candidate genes associated with bicuspid aortic valve, revealing pronounced heterogeneity and complex interplay between coding and regulatory mutations. Collectively, our findings establish rare regulatory mutations as contributors to the heritability of bicuspid aortic valve and underscore the need to elucidate their mechanistic roles in disease pathogenesis.

Bicuspid aortic valve (BAV) is the most common congenital cardiac defect, affecting 0.5–1.5% of the global population[1–3]. Unlike the standard tricuspid aortic valve (TAV), BAV forms with two instead of three cusps, predisposing individuals to severe cardiovascular complications, such as aortic valve stenosis and ascending aortic aneurysm[4].

Over half of BAV patients require surgical interventions, including valve replacement or repair, during their lifetime[5].

Despite an estimated heritability of up to 90%[6], the genetic etiology of BAV remains largely unresolved[7]. Familial studies have implicated protein-coding mutations in several genes, including

[1]Science for Life Laboratory, School of Engineering Sciences in Chemistry, Biotechnology and Health, Division of Gene Technology, KTH Royal Institute of Technology, Solna, Sweden. [2]Cardiology Unit, Center for Molecular Medicine, Department of Medicine, Karolinska Institutet, Karolinska University Hospital, Stockholm, Sweden. [3]Institute of Protein Research, Russian Academy of Sciences, Pushchino, Russia. [4]Institute of Biochemistry and Genetics, Ufa Federal Research Centre of the Russian Academy of Sciences, Ufa, Russia. [5]Life Improvement by Future Technologies (LIFT) Center, Moscow, Russia. [6]Faculty of Bioengineering and Bioinformatics, Lomonosov Moscow State University, Moscow, Russia. [7]Department of Molecular Medicine and Surgery, Karolinska Institutet, Karolinska University Hospital, Stockholm, Sweden. [8]Vavilov Institute of General Genetics, Russian Academy of Sciences, Moscow, Russia. ✉e-mail: artemii.zhigulev@ki.se; hanna.bjorck@ki.se; pelin.akan@scilifelab.se

*ADAMTS19*[8], *GATA6*[9], *NOTCH1*[10], *ROBO4*[11], *SMAD6*[12], and *TBX5*[13]. Genome-wide association studies have further identified a limited number of common regulatory variants, such as those near *GATA4*[14]. However, these combined findings account for less than 10% of cases, highlighting the need to investigate rare regulatory variants—an underexplored factor in both BAV research and human genetics more broadly[15].

Regulatory variants can influence gene expression by altering transcription factor (TF) binding sites in promoters and enhancers, thereby affecting chromatin rewiring at the single promoter–enhancer interaction level[16]. Since more than two-thirds of enhancers regulate non-adjacent genes, uncovering their molecular mechanisms requires comprehensive mapping of three-dimensional (3D) genome organization[17]. Prior studies have made important strides in this direction, but were typically limited by a focus on large-scale 3D genome changes and structural variants, or low-resolution promoter–enhancer maps[18].

BAV presents an additional challenge: valve morphogenesis occurs early in embryogenesis, but patient samples are usually limited by adults with already malformed valves. Nevertheless, accumulating evidence suggests that deleterious regulatory interactions active during embryonic development may persist in adult tissues[19,20]. Based on these premises, we hypothesized that rare regulatory mutations may drive chromatin rewiring events associated with BAV formation and that some of these pathological interactions could be captured in adult ascending aortic endothelial cells (AECs).

In this study, we combined patient-specific promoter–enhancer interactome mapping using promoter capture Hi-C (HiCap)[21] with matched whole-genome sequencing (WGS) (Fig. 1a, b). Leveraging additional datasets, this strategy allowed us to delineate the contribution of rare regulatory variants to BAV, estimate their effective minor allele frequencies, identify the most affected developmental cell states and pathways, as well as substantially expand the BAV genetic network beyond previously described contributors (Fig. 1c, d).

## Results

### HiCap efficiently captures the promoter-enhancer interactions of patient ascending aortic endothelial cells

The study included eight BAV and eight TAV patients undergoing elective open-heart surgery at Karolinska University Hospital, all with homogenous clinical profiles (Methods, Supplementary Table 1). WGS identified an average of $323,387 \pm 18,973$ short germline variants,

including single-nucleotide variants (SNVs) and indels with a minor allele frequency (MAF) below 10%. Structural variants (SVs) were also detected. We curated a *tier 1* gene set, comprising genes with a known monogenic contribution to BAV and recurrent observations in human datasets (Supplementary Data 1). Consistent with previous studies, among the *tier 1* genes, we identified a single nonconclusive moderate-impact missense mutation, 9-136496711-T-A (p.Gln2343Leu), in *NOTCH1* in patient BAV7. That underscores the likely contribution of non-coding variants to BAV pathogenesis.

Non-coding variants accounted for up to 95% of the identified variation (Supplementary Fig. 1a–c). To prioritize regulatory mutations and link them to their target genes in the relevant cellular context, we mapped promoter-anchored interactions using HiCap in AECs of each patient, explanted from the ascending aorta tissue biopsies and expanded in culture (Methods)[22]. Ascending AECs are a relevant model for studying BAV pathology due to their shared embryonic lineage with aortic valve cells[23]. HiCap targeted 23,111 promoters and 1,123 selected variants (Methods), achieving whole-genome coverage and resolution sufficient to resolve individual regulatory regions (821 bp). In total, we identified 1,264,834 unique statistically significant promoter-anchored interactions (Methods, Table 1 and Supplementary Data 2).

To validate the HiCap dataset, we analyzed promoters and their corresponding promoter-interacting regions (PIRs) separately. First, we showed that endothelial-specific genes exhibited significantly higher expression levels when involved in HiCap interactions than housekeeping genes (Fig. 2a, b and Supplementary Data 1)[24]. This finding confirms HiCap's ability to capture functional, endothelial-specific interactions in the explanted and expanded AECs, demonstrating the relevance of the model.

Second, the PIRs detected by HiCap showed a significant overlap with the classical active enhancer mark H3K27ac in human ascending AECs (for common regulatory regions shared between all patients: $p = 0$, two-tailed Fisher's exact test, 4.4% of the genome covered). HiCap PIRs also substantially overlapped with enhancer regions in the endothelial model cell line TeloHAEC (Supplementary Data 3), which shares many key characteristics with aortic endothelial cells (Fig. 2c)[25]. This supports referring to these HiCap-derived PIRs as enhancers, consistent with previous studies[21]. Notably, we also identified a subset of patient-specific enhancers, highlighting even higher inter-individual variability in the regulatory landscape than previously reported (Fig. 2c)[26,27].

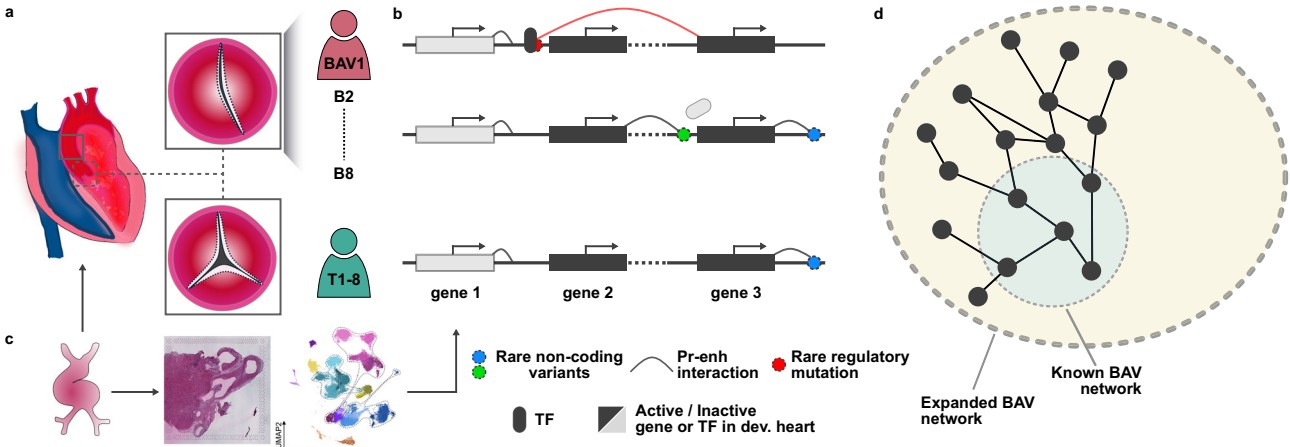

**Fig. 1 | Study overview. a** AECs were isolated from the ascending aorta (solid gray square) of sixteen adult patients with BAV or TAV (dashed gray square) to map patient-specific promoter-enhancer interactions with HiCap. **b** These interactions were integrated with matched WGS data to identify mutation-driven chromatin rewiring events, including predicted effects on TF binding affinity. **c** Developmental heart scRNA-seq and ST datasets were leveraged to provide developmental context. **d** The findings expand the genetic network of BAV, establishing an advanced resource for BAV research. AECs aortic endothelial cells, BAV or B bicuspid aortic valve, scRNA-seq single-cell RNA sequencing, ST spatial transcriptomics, TAV or T tricuspid aortic valve, TF transcription factor, WGS whole-genome sequencing.

**Table 1 | Numerical overview of the filtering strategy**

| | Features | BAV cohort specific | TAV cohort specific | Comments |
|---|---|---|---|---|
| **HiCap** | Interactions | 569,753 | 328,372 | We detected 1.7-fold more interactions in the BAV cohort compared to the TAV cohort. |
| | Genes | 17,217 | 15,811 | |
| | Enhancers | 484,497 | 297,279 | |
| **WGS** | Promoter mutations | 20,854 | 19,174 | No statistically significant difference in the number of promoter and enhancer mutations. |
| | Enhancer mutations | 286,413 | 268,240 | |
| **HiCap x WGS** | Concordant (1,2) | | | Concordant mutations are enriched in the BAV cohort (p-value < 2.2e-16, OR = 1.1, two-tailed Fisher's exact test). |
| | Promoter events | 58,534 | 30,131 | |
| | Enhancer events | 21,425 | 21,472 | |
| | Discordant (3) | | | |
| | Promoter events | 1,537,785 | 907,161 | |
| | Enhancer events | 625,639 | 363,362 | |

(1) 373 loss-of-interaction / 79,586 gain-of-interaction for BAV; 91 / 42,512, respectively for TAV $p < 2.2e-16$, OR = 2.4, two-tailed Fisher's exact test. (2) 851 concordant *tier 1-2* genes / 7,839 concordant housekeeping genes for BAV; 230 / 3,853, respectively, for TAV, $p = 4.8e-16$, OR = 1.819, two-tailed Fisher's exact test. (3) 15,355 *tier 1-2* genes / 228,717 housekeeping genes in BAV; 8,587 / 116,859, respectively, for TAV, $p < 1.1e-10$, OR = 0.913, two-tailed Fisher's exact test.

## Mutation-driven chromatin rewiring events affect aortic valve development genes in BAV patients

To address non-coding variants as regulatory mutations, they should be functional. Rare mutations are typically heterozygous and can act in an allele-specific way, affecting promoter-enhancer interactions on a single allele only, revealed by the allelic imbalance of read counts (Fig. 2d)[28].

We focused on the cumulative effect of the enhancer variants with potentially small individual contributions to the activity of a single promoter. Direct allele-specific analysis of HiCap data, performed without integrating WGS, did not detect any cohort-specific signal in *tier 1* genes (Supplementary Data 4). However, several *tier 2* genes with less genetic evidence linking them to human BAV disease (*EDNRA*, *PUF60*, *SNIP1*, *SMC3*) or connected to aortic valve development (*NFATC1*, *TBX20*, *TNFRSF1A*, *ZMPSTE24*) exhibited signals present exclusively in BAV patients (Fig. 2e). Notably, *ELN* was the only gene exhibiting allelic imbalance signals specific to TAV patients. Overall, genes with allelically-imbalanced interactions were overrepresented in *tier 2* set in the BAV cohort (7.1% in BAV vs. 0.9% in TAV, $p = 0.035$, OR = 8.471, two-tailed Fisher's exact test). Chromatin events involving allele-specific variants were previously shown to be tightly linked with the variance coverage and effect size[28]. Thus, these findings not only support the integration of HiCap with WGS at the next step as an effective strategy for uncovering the regulatory effects of non-coding variants but also highlight the opportunity to expand the genetic network of BAV.

Regulatory mutations can influence gene expression by either fine-tuning or altering promoter-enhancer interactions. In the case of BAV, addressing the activity fine-tuning of developmental interactions based on adult models is challenging. However, previous studies suggested that some developmental interactions may persist into adulthood[19,20]. Thus, we focused on two chromatin rewiring scenarios, where the regulatory mutation may cause either gain-of-interaction (GoI) or loss-of-interaction (LoI). We filtered the events where either component of the interaction (promoter or enhancer) overlapped with variants detected in patients. In total, we identified 53,785 promoter and 745,555 enhancer variants overlapping interacting regions (MAF < 10%, Table 1). All the events were classified into two categories (Fig. 2f):

1. Concordant events – the promoter-enhancer interaction was rewired exclusively in patients with the same variant in the corresponding promoter or enhancer.
2. Discordant events – the promoter-enhancer interaction was rewired non-exclusively in patients with the same variant in the corresponding promoter or enhancer.

We further categorized concordant and discordant events by BAV and TAV cohorts (Table 1). Overall, while the GoI scenario dominated (99.6%), the BAV-cohort had significantly more concordant LoI events than the TAV-cohort (0.47% vs 0.21%). Downstream analyses were conducted in parallel for BAV- / TAV-cohort specific and concordant/discordant interactions. Notably, while BAV-specific concordant events were 1.8-fold more prevalent in *tier 1* and *tier 2* genes than in housekeeping genes, BAV-specific discordant interactions, in contrast, showed a minimal 0.9-fold depletion. Overall, these results emphasize the predictive value of concordance filtering for regulatory mutation prioritization and BAV pathogenesis studies in particular.

## Rare regulatory mutations disrupt the transcriptomic landscape of fetal heart mesenchymal cells in BAV patients

We next investigated how rare regulatory mutations contribute to chromatin rewiring in BAV patients. Because functional consequences are expected to occur indirectly through altered TF binding, we annotated prioritized variants with non-redundant TF motifs from HOCOMOCO (Supplementary Data 5 and Fig. 3a)[29]. As with chromatin interactions, two primary classes were considered: gain-of-motif (GoM, 46.5%) and loss-of-motif (LoM, 53.5%) (Fig. 3b). Notably, binding site alterations involving the *tier 2* NFAT-related TF family, including *NFATC1*, which was also implicated in our allele-specific analysis, were significantly overrepresented in the BAV compared with the TAV cohort (0.16% for BAV, 0.06% for TAV, FDR < 2.6e-47, OR = 2.647, two-tailed Fisher's exact test). NFAT-related TFs are key regulators of endocardial cushion development and endothelial cell fate, highlighting their potential role in BAV pathogenesis[30].

To derive a cell-type–specific framework of aortic valve development and prioritize concordant events potentially active during embryogenesis, we integrated scRNA-seq and spatial transcriptomics (ST) data spanning 5.5–14 weeks post-conception[31]. This dataset comprises 31 coarse-grained cell types and 72 fine-grained states, enabling functional interpretation of regulatory mutations in developmental contexts.

Given the lack of prior knowledge, we applied a data-driven approach agnostic to the cell type, assuming that the relevant cell types would be enriched for altered transcripts. We defined a transcript as altered if its promoter participated in a concordant event and both the transcript itself and the TF affected by the corresponding variant were expressed (Fig. 3a). Variants were stratified by minor allele frequency (MAF): rare (MAFa, 0–2%), moderately rare (MAFb, 2–4%), and common (MAFc, 5%). Promoter and enhancer variants were analyzed separately (Methods).

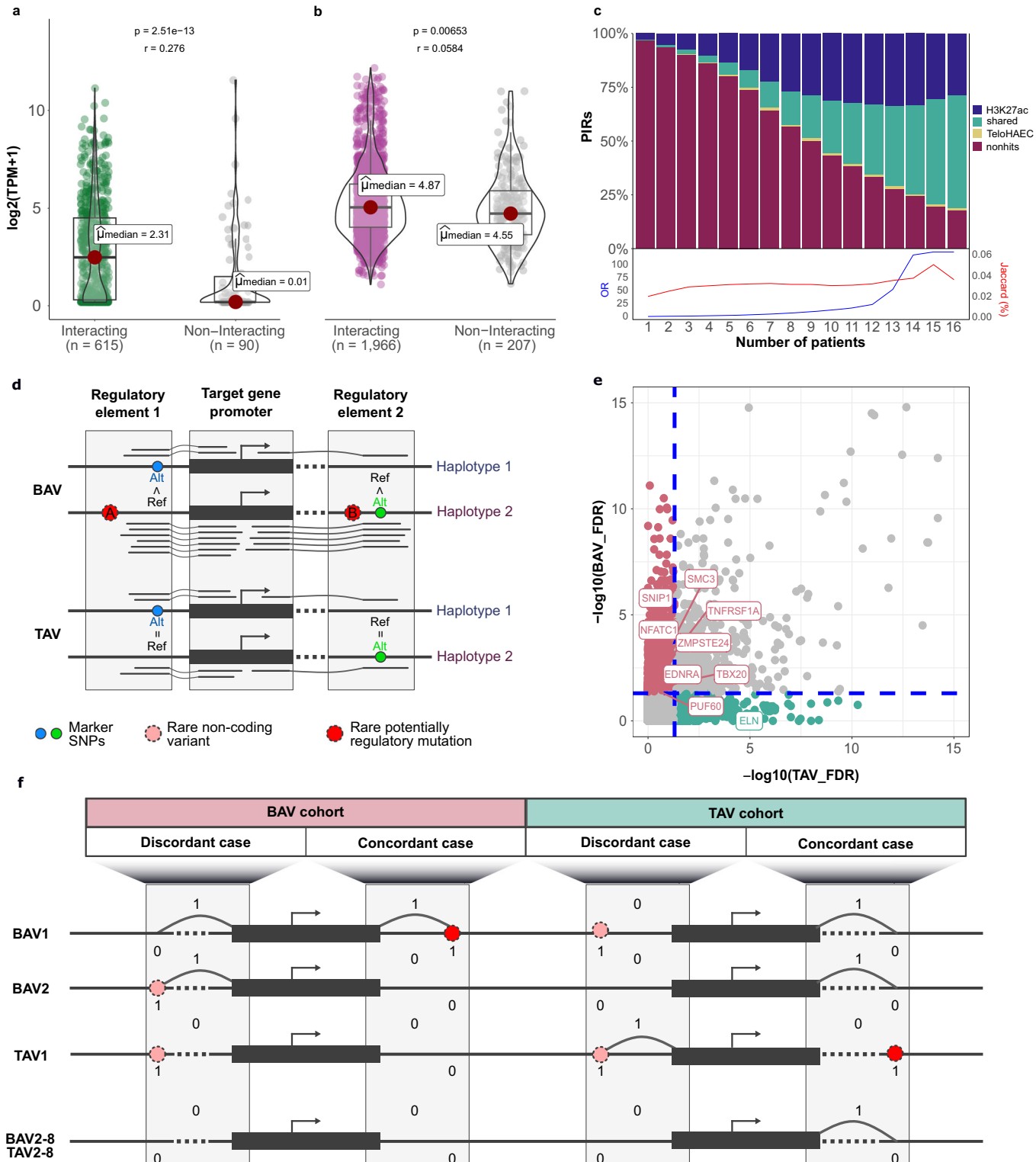

**Fig. 2 | Interactome analysis and filtering strategy.** Comparison of expression levels of (**a**) endothelial-specific and (**b**) housekeeping genes involved into HiCap interactions or not. Data are shown as individual data points. Central tendency is represented by the median, and error bars indicate the 95% confidence interval around the median. Statistical significance was assessed using a two-sided Wilcoxon rank-sum test. Effect size (r) is reported. Source data are provided as a Source Data file. **c** Overlap of HiCap-derived promoter interacting regions with H3K27ac ChIP-Seq peaks from different patients and cumulative TeloHAEC enhancer regions. The Odds Ratio between HiCap and H3K27ac regions is in blue, while the Jaccard similarity index is in red. **d** Schematic of allele-specific interaction analysis. Arches represent chimeric reads spanning the ligation sites, A and B correspond to the different regulatory mutation locations. **e** Per-gene aggregation of allele-specific interaction enrichments in BAV and TAV cohorts. FDR thresholds of 0.05 (blue dashed lines) are shown for TAV and BAV values. **f** Representation of BAV- / TAV-cohort specific, concordant/discordant and gain-of-interaction/lose-of-interaction events. BAV bicuspid aortic valve, HiCap sequence capture HiC, TAV tricuspid aortic valve, TPM transcript per million.

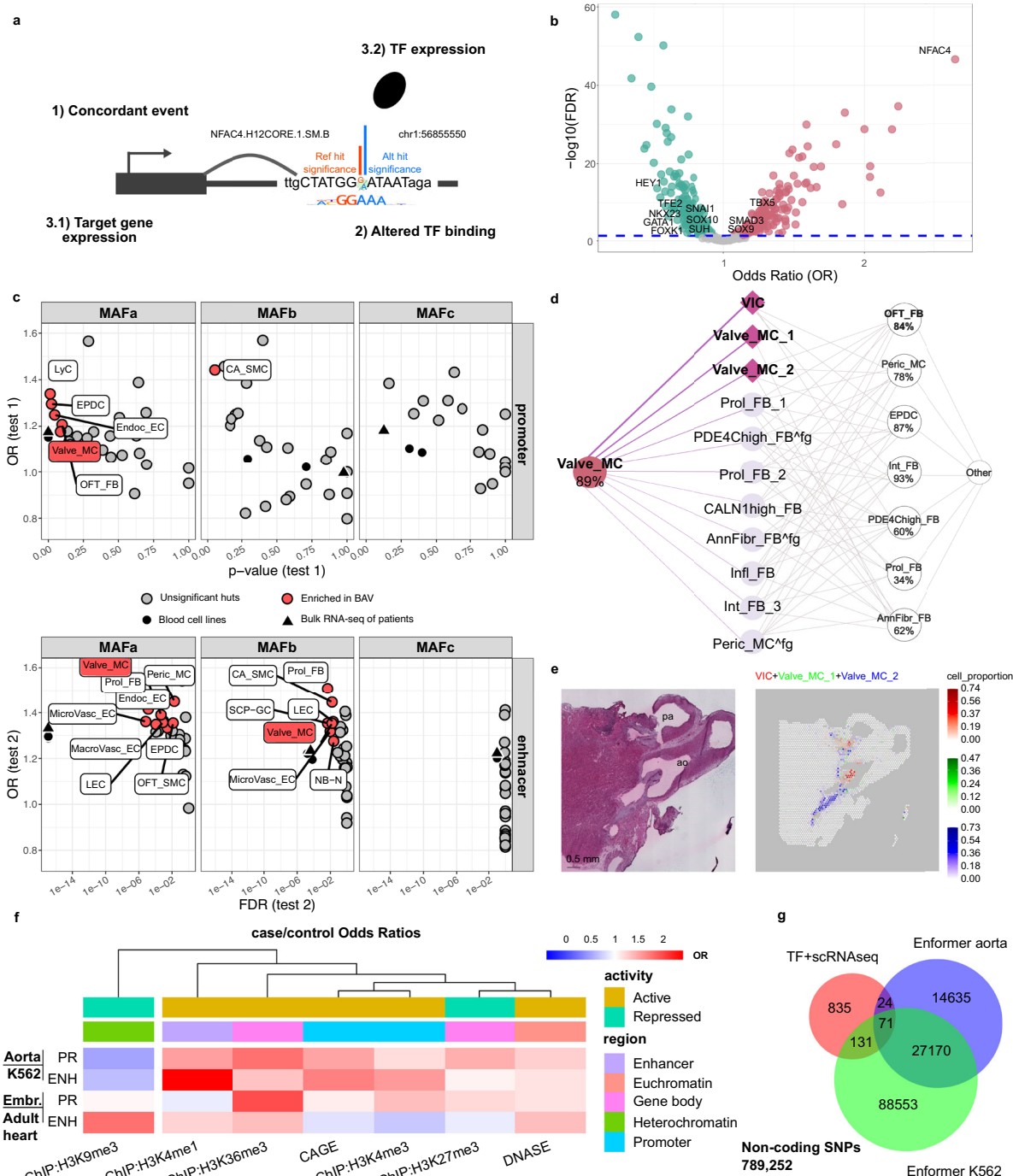

**Fig. 3 | Cellular insights into BAV etiology. a** Schematic of the active concordant events filtering strategy. **b** Most frequently affected TF families in BAV (red) and TAV (green) cohorts. Source data are provided as a Source Data file. **c** Impact of promoter and enhancer concordant mutations across different MAF ranges on developmental cell types (two-tailed Fisher's exact test). Promoter (test 1) and enhancer (test 2) mutations were analyzed separately. DE genes between cell types were included (Wilcoxon rank-sum test). Source data are provided as a Source Data file. **d** Contribution of different cell states to Valve_MC. The top 3 contributors are highlighted. **e** Spatial mapping of affected cell states (proportion per Visium spot): VIC (red), Valve_MC_1 & Valve_MC_2 (green, combined). Results are shown together to illustrate spatial relationships. **f** Top rows: enrichment of ascending aorta regulatory variants (predicted by Enformer) in the prioritized variant set compared to all promoter/enhancer-overlapping variants. K562 tracks served as control. Bottom rows: enrichment of embryonic heart regulatory variants in the prioritized variant set compared to all promoter/enhancer-overlapping variants. Adult heart tracks served as control. **g** Venn diagram of variants prioritized by concordancy, TF

affinity changes, scRNA-seq (red), and Enformer-predicted regulatory mutations based on DNase tracks for aorta (blue) and K562 (green). AnnFibr_FB annulus fibrosus fibroblasts, BAV bicuspid aortic valve, CA_SMC large coronary artery smooth muscle cells, CALN1high_FB CALN1-enriched fibroblast population, DE differentially expressed, Endoc_EC endocardial cells, EPDC epicardium-derived progenitor cells, fg fine-grained, Infl_FB inflammatory mediator-enriched fibroblasts, Int_FB interstitial fibroblasts, LEC lymphatic endothelial cells, LyC lymphoid cells, MacroVasc_EC macrovascular endothelial cells, MAF minor allele frequency, MicroVasc_EC microvascular endothelial cells, NB-N neuroblasts and neurons, OFT_FB fibroblasts around the outflow tract and developing great arteries, OFT_SMC smooth muscle cells near outflow tract and great arteries, PDE4Chigh_FB PDE4C-enriched fibroblasts, Peric_MC pericyte-like mesenchymal cells, Prol_FB proliferating fibroblasts, scRNA-seq single-cell RNA sequencing, SCPGC Schwann cell progenitors and glial cells, ST spatial transcriptomics, TAV tricuspid aortic valve, Valve_MC cardiac valve-related mesenchymal cells, VIC valve interstitial cells.

In BAV patients, cardiac valve–associated mesenchymal cells (Valve_MC) was the most affected population (i.e., contained most affected transcripts as described above), showing significant enrichments for promoter ($p < 0.1$, Fisher's exact test) and enhancer (FDR < 0.05, Fisher's exact test) mutations (Fig. 3c). That signal was concentrated in the MAFa range for promoter variants, while effects of enhancer variants extended into MAFb, consistent with the higher average impact expected for promoter mutations[32]. Fine-grained scRNA-seq further revealed that the primary sources of the Valve_MC cell type were valve interstitial cells (VIC) and Valve_MC states 1 and 2 (Fig. 3d). ST confirmed their proximity to the developing aortic valve, strengthening their relevance to BAV pathogenesis (Fig. 3e). In contrast, TAV samples displayed only weak transcriptomic effects.

To further evaluate our prioritization strategy, we annotated variants using the Enformer deep learning model (Supplementary Data 6)[33]. Variants involved in concordant interactions active in VIC and Valve_MC states 1/2 were more frequently predicted to have a regulatory impact in the adult ascending aorta than all non-coding promoter/enhancer variants (Fig. 3f, top). Similarly, prioritized variants were more often predicted to be regulatory in embryonic rather than adult heart tissues (Fig. 3f, bottom). As expected, euchromatin (DNase) and heterochromatin (H3K9me3) marks showed generally opposite effects. Although Enformer predicted approximately 40-fold more regulatory mutations based on DNase tracks (Fig. 3g), the consistency across approaches supports the robustness of our variant prioritization strategy.

Together, these results demonstrate that, in BAV patients exclusively, rare and moderately rare regulatory mutations reshape the transcriptomic landscapes of fetal heart mesenchymal populations critical for aortic valve development, underscoring the value of cell-type–specific filtering for non-coding variant prioritization.

### Endocardial cushion molecular programs are disrupted in mesenchymal cells of BAV patients

Based on our analysis, we compiled a final list of 198 BAV-specific case genes (approximately thirty times more than previously described) and 112 TAV-specific control genes affected by concordant chromatin rewiring events active in VIC and Valve_MC 1/2 cell states during aortic valve formation (Supplementary Data 1). Notably, 20 BAV-specific genes were involved in allele-specific interactions, with no TAV-specific matches. To deepen the biological interpretation of the expanded BAV genetic pathway, we next examined which molecular pathways might be perturbed by dysregulation of these genes. Alongside genes affected by chromatin rewiring events active in VIC and Valve_MC 1/2, we also assessed genes affected in other spatially relevant populations, including valve endothelial cells (inflow, IF_VEC; outflow, OF_VEC), arterial endothelial cells (Art_EC_1), and ventricular cardiomyocytes (vCM_1) (Supplementary Fig. 2)[31].

Concordant BAV events active in mesenchymal cell states showed significant enrichment for tier 1 and tier 2 genes and pathways associated with valve development and cushion formation, particularly transforming growth factor beta (TGFβ) signaling and epithelial-to-mesenchymal transition (EMT) (Fig. 4a). Tier 1 and tier 2 genes exhibited a similar but more spatially sparse enrichment pattern within the aortic valve region compared to EMT, whereas TGFβ-related genes displayed a distinct spatial distribution in the same area (Fig. 4b). BAV-specific concordant events involving TGFβ and EMT pathways are detailed in Table 2. Notably, one such example is BMP2, a key regulator of EMT during endocardial cushion formation that later remodels into the cardiac valves (Fig. 4c)[34]. In contrast, discordant events showed limited and less specific gene set enrichment patterns (Supplementary Fig. 3).

Together, these findings strongly link the affected cell states to specific biological processes, indicating that disruption of endocardial cushion programs within mesenchymal cells is a central feature of BAV.

They also suggest that previous models of BAV genetics, largely focused on protein-coding and common regulatory variants, capture only a fraction of the disease's underlying biology.

While mesenchymal cell states exhibited strong and consistent gene set enrichments, spatially relevant endothelial cell states showed only limited signals, and cardiomyocyte states were minimally affected by concordant events. Interestingly, the predominant signal in adult ascending AECs did not originate from the positionally and biologically closest developmental cell state, Art_EC_1, but rather from mesenchymal cell states. A heatmap of prioritized tier 1 and tier 2 genes reinforced this trend (Fig. 4d). Collectively, these broad and partly overlapping enrichment patterns point to substantial heterogeneity in BAV pathology.

### Newly implicated genes in BAV etiology highlight its high heterogeneity

When examining the genetic architecture of BAV, we found that each individual carried numerous rare and often unique regulatory mutations influencing dozens of genes. Applying our prioritization framework enabled the progressive refinement of relevant, concordant, and functionally active regulatory mutations (Fig. 5a). Although individual regulatory mutations may exert modest effects, their cumulative impact, or interaction with high-impact protein-coding mutations, can be substantial. Notably, approximately 20% of the genes prioritized from HiCap data carried both mutation classes, and all patients exhibited a mixture of coding and regulatory mutations (Fig. 5b). Despite the rarity and individuality of regulatory mutations, approximately 10% of the affected genes were shared among multiple patients, underscoring the marked genetic heterogeneity of BAV pathology (Fig. 5c).

To further explore this heterogeneity, we constructed protein-protein interaction (PPI) networks for the active concordantly affected, cohort-specific genes, including the tier 1 genes for the BAV cohort ("Methods"). The BAV-specific network (nodes = 200, edges = 273, average node degree = 2.73, PPI enrichment $p < 1.0$e-16) exhibited substantially more interactions than the TAV-specific network (nodes = 103, edges = 74, average node degree = 1.44, PPI enrichment $p = 1$e-07) (Fig. 5d). These networks reveal that the prioritized BAV gene set is non-random and forms a coherent cellular program potentially contributing to valve malformation during embryogenesis.

Together, these findings demonstrate that rare regulatory mutations contribute significantly to BAV formation by disrupting a defined subset of genes involved in mesenchymal molecular pathways and endocardial cushion development. Coding and regulatory variants often act synergistically, sometimes within the same gene, reflecting the complexity and heterogeneity underlying BAV etiology.

## Discussion

This study presents an integrative, whole-genome interactome analysis comparing patients with bicuspid (BAV) and tricuspid (TAV) aortic valves, combining promoter-enhancer interaction mapping (HiCap) with matched whole-genome sequencing. In contrast to previous genome-wide association studies, which primarily focused on common regulatory mutations and assigned them to the nearest genes, our approach enabled the identification of all regulatory mutations, irrespective of minor allele frequency, and their precise mapping to cognate target genes.

Our in silico functional analyses revealed that genes previously associated with BAV were not enriched for allele-specific interactions. Integration with developmental spatial transcriptomic data further revealed that BAV-related genes and aortic valve developmental genes impacted by rare mutation-driven chromatin rewiring exhibited spatial enrichment patterns similar to those of EMT genes affected by mutations, and distinct from the mutation-affected genes within the TGFB pathway. These findings suggest that earlier genetic models of BAV,

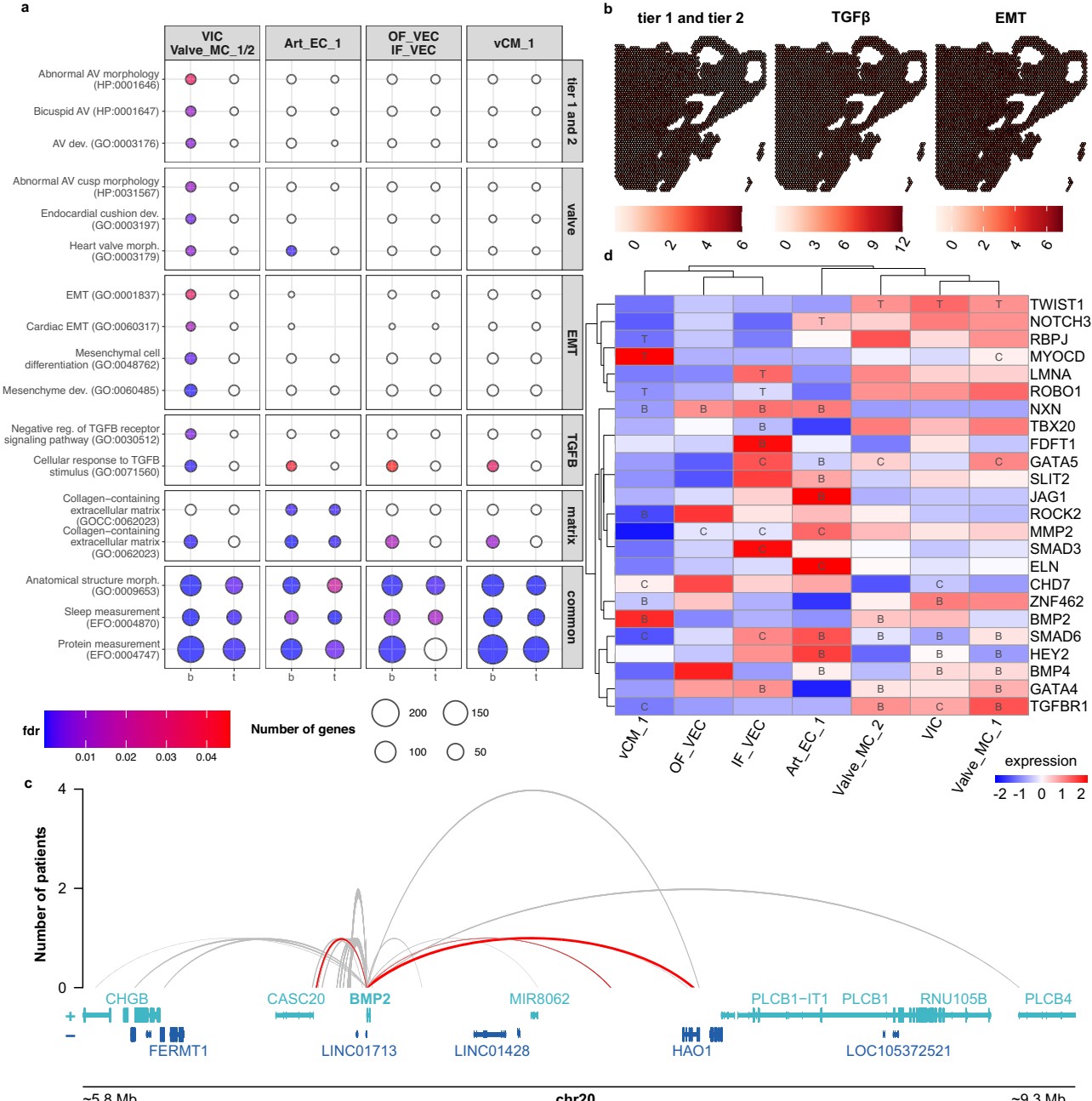

**Fig. 4 | Molecular pathways insights into BAV etiology. a** Gene set enrichment analysis of genes affected by concordant events active in different spatially prioritized cell states. DE genes were used. **b** Spatial binary mapping of concordantly affected genes (BAV vs. TAV) within key gene sets. **c** Example of the promoter-enhancer interactome profile of the *BMP2* with active concordant events (red) and other interactions (gray). **d** Gene-level analysis of *tier 1* and *tier 2* genes affected by concordant events in BAV (B), TAV (T), and shared (common - C) cohorts, linked to overall expression levels. Hierarchical clustering was performed based on the B (3), T (-3), C (0), NA (0) matrix. DE genes were analyzed. Art_EC_1 arterial endothelial cells, BAV bicuspid aortic valve, DE differentially expressed, EMT epithelial-tome-senchymal transition, IF_VEC inflow valve endothelial cells, OF_VEC outflow valve endothelial cells, TAV tricuspid aortic valve, TGFβ transforming growth factor beta, VIC valve interstitial cells, Valve_MC cardiac valve-related mesenchymal cells, vCM_1 ventricular cardiomyocytes.

primarily based on protein-coding or common regulatory variants, may have overlooked important regulatory disruptions.

Using an agnostic, data-driven framework guided by developmental single-cell RNA-seq data, we identified approximately thirty times more genes associated with BAV than previously reported. These genes play key roles in endocardial cushion formation and are differentially expressed in embryonic mesenchymal cells spatially related to the aortic valve region. While some are recurrently affected across patients, our results indicate a convergent mechanism in which the synergistic effects of multiple unique coding and non-coding mutations

collectively disrupt valve development. A particularly notable finding is the systematic involvement of the NFAT-related TF family. Although *NFATC1* itself exhibited allele-specific promoter-enhancer interactions, alterations in NFAT binding motifs were significantly overrepresented across the broader NFAT TF family in BAV patients.

Two key insights emerge from these findings. First, we observed a striking persistence of developmental regulatory interactions in adult endothelial cells. Although consistent with prior reports, our data provide a systematic, genome-wide demonstration of this phenomenon[19,20]. The extent to which developmental, either

**Table 2 | BAV-specific genes affected in TGFβ and EMT pathways**

| Gene | Patient ID | SNP general | | | | | Enh/Pr | SNP/ Indel | Motif |
|---|---|---|---|---|---|---|---|---|---|
| | | Chr | Start | End | REF | ALT | | | |
| BMP2 | 10000000 | chr20 | 7628203 | 7628204 | G | A | enh | snp | CUX1 |
| CAV1 | 10000000 | chr7 | 116266255 | 116266256 | G | A | enh | snp | ERG, ETV1, ELF1 |
| CHST11 | 10000000 | chr12 | 104453118 | 104453119 | G | C | enh | snp | JUN, JUND, BACH2, FOS, FOSB, FOSL2, ATF3, PBX3 |
| FOS | 10000000 | chr14 | 75870402 | 75870403 | C | T | enh | snp | HAND2 |
| LDLRAD4 | 10000000 | chr18 | 13150865 | 13150866 | C | A | enh | snp | AR, JUN, JUND, BACH2, FOS, FOSB, FOSL2, ATF3, MEF2C, MEIS2 |
| LTBP2 | 10000000 | chr14 | 74606930 | 74606931 | C | T | enh | snp | TBX20, TBX5 |
| RASL11B | 10000000 | chr4 | 52328819 | 52328820 | T | C | enh | snp | HEY2 |
| RASL11B | 10000000 | chr4 | 52521660 | 52521661 | C | CTAAAA | enh | indel | |
| SEMA3D | 10000000 | chr7 | 85186695 | 85186696 | T | C | pr | snp | AR, THRB, SALL1, ZNF124 |
| SMAD6 | 10000000 | chr15 | 66629879 | 66629880 | G | A | enh | snp | GLIS1, NFIB, NFIC |
| LDLRAD4 | 01000000 | chr18 | 13218463 | 13218464 | C | T | pr | snp | EGR1 |
| LDLRAD4 | 01000000 | chr18 | 13221311 | 13221312 | T | C | enh | snp | SALL1 |
| LDLRAD4 | 01000000 | chr18 | 13286895 | 13286896 | A | G | enh | snp | GLI3, HAND2, SNAI1, TCF12, TWIST1 |
| FUT8 | 00100000 | chr14 | 65409709 | 65409710 | C | T | pr | snp | ZNF423 |
| TBX3 | 00100000 | chr12 | 114674638 | 114674639 | C | A | enh | snp | BACH2 |
| BMP4 | 00010000 | chr14 | 54312707 | 54312708 | C | G | enh | snp | AR |
| FOS | 00010000 | chr14 | 75660741 | 75660742 | C | G | enh | snp | HEY2 |
| HEY2 | 00010000 | chr6 | 125983543 | 125983544 | A | G | enh | snp | CREM, NFIB, NFIC |
| PDGFD | 00010000 | chr11 | 103906264 | 103906265 | C | T | enh | snp | RFX7, TCF12, TWIST1 |
| SFRP1 | 00001000 | chr8 | 41200733 | 41200734 | T | TAGA | enh | indel | |
| SFRP1 | 00001000 | chr8 | 41200734 | 41200735 | C | G | enh | snp | MEF2A, MEF2C |
| SFRP1 | 00001000 | chr8 | 41200736 | 41200737 | A | AT | enh | indel | |
| SFRP1 | 00001000 | chr8 | 41202307 | 41202308 | C | T | enh | snp | SALL1 |
| BMP2 | 00000010 | chr20 | 6589795 | 6589796 | C | T | enh | snp | EGR1 |
| BMP2 | 00000010 | chr20 | 6768533 | 6768534 | CTGT | C | pr | indel | |
| GATA4 | 00000010 | chr8 | 11410795 | 11410796 | G | A | enh | snp | GLIS1 |
| SCX | 00000010 | chr8 | 144522234 | 144522235 | C | A | enh | snp | GLI3 |
| SMAD7 | 00000010 | chr18 | 48563268 | 48563269 | TG | T | enh | indel | |
| SMAD7 | 00000010 | chr18 | 48565942 | 48565943 | AGAGT | A | enh | indel | |
| ZFP36L1 | 00000010 | chr14 | 68475651 | 68475652 | CT | C | enh | indel | |

physiological or pathogenic, chromatin interactions persist into adulthood remains unclear, as do the underlying mechanisms that govern the rules of the chromatin "recoloring" dynamics. Nevertheless, even under a simplified model of gain- or loss-of-interaction events, the use of multiple orthogonal controls − BAV versus TAV comparisons, concordant versus discordant events, common versus rare variants, and aortic tissue versus blood-derived transcriptomes − enabled us to robustly link rare regulatory mutations to mesenchymal dysregulation and impaired endocardial cushion formation.

Second, rare regulatory mutations identified from adult endothelial datasets were more frequently associated with genes expressed in mesenchymal rather than endothelial or cardiomyocyte lineages. This observation suggests a potential lineage-passing mechanism, whereby altered developmental regulatory interactions, influencing non-endothelial valve cell types, persist into adulthood. Whether adult valve endothelial, interstitial, or aortic smooth muscle cells display distinct or overlapping developmental regulatory signatures remains an open question and represents a key avenue for future investigation.

Although our results may not capture the full genetic spectrum of BAV, they provide critical mechanistic insights that advance understanding of its formation. The main limitation arises from patient selection: only individuals with ascending aortic dilatation were included, as endothelial cells could not be isolated from the small biopsies of non-dilated aortas. In addition, most analyzed BAV patients

exhibited the left−right cusp fusion pattern, and the overall cohort size was constrained by the practical challenges of obtaining sufficient endothelial tissue samples.

Because HiCap requires approximately one million cells per sample, even dilated biopsies yielded insufficient material, necessitating in vitro expansion of endothelial cells under static conditions. Although this system does not fully reproduce the biomechanical environment of the native aortic endothelium, the cultured cells retained key endothelial features, and their regulatory landscapes remained lineage-specific. While some chromatin rewiring may arise during culture, the identical handling of BAV and TAV samples ensures that observed differences reflect genuine biological signals rather than technical artifacts. Notably, new HiCap protocols now require fewer than 0.5 million cells, a development that will facilitate future studies using smaller patient-derived biopsies.

Annotating regulatory mutations through TF motif analysis proved essential for non-coding variant prioritization. Although HOCOMOCO v13 encompasses nearly 70% of human TFs, its extensive motif diversity and the burden of multiple testing result in many variants appearing to alter at least one motif. This limits the interpretive power of motif-based filtering alone. Therefore, prioritization of both target genes and relevant TFs is critical. We addressed this limitation through integration with developmental scRNA-seq data, although TFs remain underrepresented in RNA-seq owing to their low expression

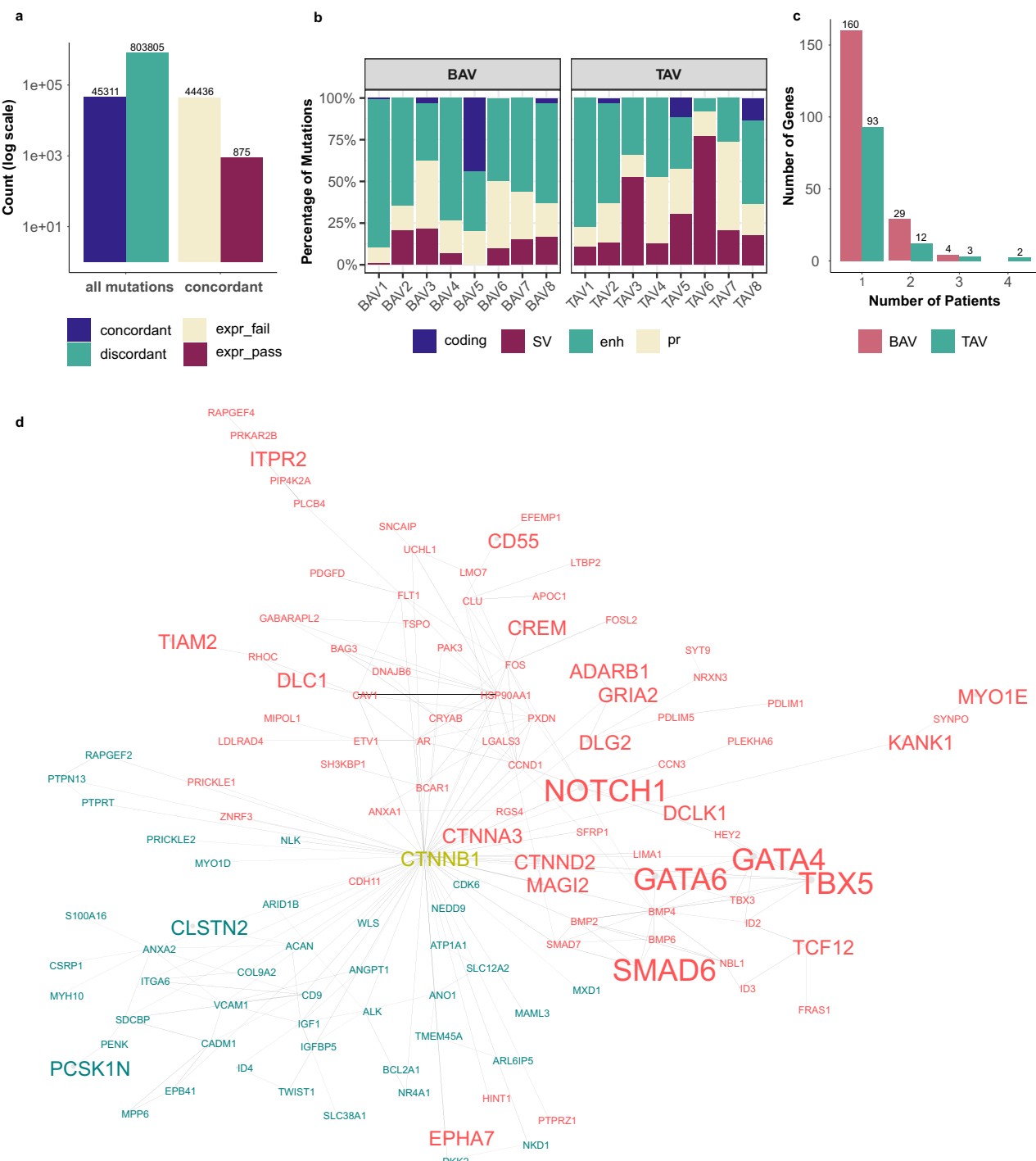

**Fig. 5 | Extended BAV pathway network analysis. a** Non-coding variant prioritization steps. **b** Distribution of different mutation types within prioritized genes across patients. **c** Distribution of common genes across patients. **d** PPI networks of BAV-specific (red) and TAV-specific (green) genes, as well as *tier 1* genes. Isolated nodes are omitted for clarity. Node size reflects if the gene is *tier 1* (large), have protein-coding (medium) or regulatory mutation (small). Edge thickness indicates STRINGdb interaction confidence scores. Gene in yellow represents an additional interactor. BAV bicuspid aortic valve, MAF minor allele frequency, PPI protein-protein interaction, TAV tricuspid aortic valve.

levels. Complementary approaches, such as Motif Activity Response Analysis, could further refine TF associations but require additional datasets capable of pinpointing patient-specific promoter activity.

Finally, the human promoter-enhancer interactome generated here represents a valuable resource for identifying regulatory alterations underlying other cardiovascular disease-related changes[35–38].

In summary, this work advances the understanding of BAV formation and BAV-associated aortopathy, reinforcing the view that BAV has a substantial genetic component. However, the genetic component is very complex and heterogenic, which may have implications for future screening strategies, including the identification of targets for familial genetic testing.

# Methods

## Patient cohorts

In this study, we included eight BAV and eight TAV individuals with ascending aortic dilatation undergoing elective open-heart surgery for ascending aortic dilatation (diameter > 40 mm), with or without concomitant aortic valve replacement, at the Cardiothoracic Surgery Unit, Karolinska University Hospital, Stockholm, Sweden. None of the individuals had significant coronary artery disease, based on coronary angiography (The ASAP/DAVAACA study). The gender distribution was 7 males / 9 females, age varied from 47 to 80 years with the average of 66 years, and all patients were Caucasian except one. Participation in this project is entirely voluntary and does not provide financial compensation.

## Whole genome sequencing and variant calling

Patient blood samples were collected at baseline before treatment started. The Blood & Cell Culture DNA Midi Kit (Qiagen) was used according to the manufacturer's protocol to extract DNA from samples. Sequencing libraries were then prepared with the KAPA HTP Library Preparation Kit (Kapa Biosystems) with a PCR-free workflow, according to the manufacturer's protocol. Samples were sequenced on NovaSeq6000 (NovaSeq Control Software 1.7.5/RTA v3.4.4) with a 101nt(Read1)-8nt(Index1)-8nt(Index2)-101nt(Read2) setup using 'NovaSeqXp' workflow in 'S4' mode flowcell. The Bcl to FastQ conversion was performed using bcl2fastq_v2.20.0.422 from the CASAVA software suite. The quality scale used is Sanger / phred33 / Illumina 1.8 + .

FastQ files were then processed using the nf-core/sarek/2.7.1 pipeline[39-41] (Supplementary Table 2). Briefly, sequences were aligned to GRCh38 (BWA)[42] and preprocessed based on GATK4 Best Practices (GATK/4.1.7.0). SNPs and small indels were called using GATK HaplotypeCaller[43], recalibrated according to GATK Variant Quality Score Recalibration algorithm, and annotated with snpEff/4.3t[44]. Non-coding effects included the following annotations: intron_variant, intergenic_region, downstream_gene_variant, upstream_gene_variant, intragenic_variant, TF_binding_site_variant, TFBS_ablation. SweGen/20190204[45] was used for initial filtering, multi-allelic variants were excluded. Structural Variants were called using Manta[46] (nf-core/sarek pipeline v3.1.2, Manta v. 1.6.0) and annotated with ensemblVEP/106.1[47] and snpEff/5.1 d[44].

## Tier 1 and tier 2 gene sets establishment

Tier 1 gene set was curated based on the monogenic contribution to BAV and recurrent observations in human datasets. Tier 2 gene set, comprising genes previously associated with BAV (HP:0001647, DOID:0080332) and genes associated with more general aortic valve terms (GO:0003176, HP:0001646), were derived from the STRINGdb v12[48] (R package v2.12.1).

## Analysis of protein-coding mutations

High and moderate-impact mutations were included in the analysis. Short variants were additionally annotated with GnomAD v4[49] with the overall MAFmax <0.001. For moderate impact short variants, changes in hydropathy, charge and polarity of amino acids were used for prioritization. All high and moderate-impact SVs were included.

## Aortic biopsy collection

Aortic biopsies were collected directly in the operating theater and washed in PBS containing Calcium chloride and Magnesium chloride. The medial layer was separated from the adventitia using adventectomy and subjected to enzymatic digestion using a solution of 1 mg/mL collagenase A (Roche, 11088793001) in dispase 1 U/mL (Stemcell Technologies, 07923) for 20 to 25 min at 37 °C, with regular gentle rocking of the Petri dish. After incubation, the endothelial side of the tissue was carefully scraped 5 to 7 times using a sterile scalpel. The collagenase solution containing endothelial cells was collected in a tube, and the tissue was rinsed a couple of times with PBS and collected in the same tube. The solution was then strained using a 100μm cell strainer and centrifuged at 400 × g for 5 min. Finally, the pellet was resuspended in 2.5 mL of EBM-2 basal medium supplemented with EGM-2 BulletKit (Lonza, CC−3162) and dispensed in a 12.5 cm2 flask previously coated with 0.2% bovine gelatin type B (Sigma, G1393). The next day, cells were gently washed twice with PBS, and a fresh endothelial cell growth medium was added. Hereafter, the medium was replaced every 2 to 3 days. Upon confluence, cells were transferred to a 75 cm2 flask and frozen at subconfluency in a solution of 90% FBS + 10% DMSO or used no later than P4. The primary endothelial cells were purified with CD31 beads (Miltenyi Biotec, #130-091-935) in MS columns (Miltenyi Biotec, #130-042-201) before use.

## HiCap and analysis

We designed 49,927 probes targeting 23,111 promoters (of 19,469 genes), 1123 GWAS variants associated with cardiovascular disease and lipid traits and 2408 negative control regions with no known regulatory activity (Supplementary Data 7).

HiCap was performed on 2,5 million cells as previously explained[22]. This method was developed by us[21] and others and provides high-resolution interaction data over genomic regions by hybridizing Hi-C material to probes targeting certain regions of interest, enabling the study of individual promoter-enhancer interactions. Briefly, the method starts with cross-linking DNA-protein-DNA complexes with formaldehyde, followed by roughly cutting DNA across the genome into ~700 bp fragments using restriction endonucleases DpnII. Spatially close fragments are then ligated before capturing promoter-enhancer sequences using probes located in known genes' promoters or probes containing selected SNVs associated with cardiovascular disease and lipid traits. Captured libraries were sequenced using HiSeq 2500 (Illumina) with HiSeq Rapid SBS Kit v2 chemistry and a 1 × 80 setup at Science for Life Laboratory (SciLifeLab, Stockholm, Sweden).

Lastly, sequencing data was analyzed for significant interactions. We mapped HiCap libraries using bwa-mem2 (version 2.2.1-20211213-edc703f)[42] and PairTools (version 1.0.2)[50]. In total, we processed 15.56 billion reads, of which 11.25 billion were uniquely mapped. Read statistics are found in Supplementary Data 8. HiCapTools v1.3.2[51] was used to call interactions in all samples. We required five or more pairs supporting each interaction and a Bonferroni-adjusted P-value < 0.05. We further filtered out trans interactions, long-range interactions with an interaction distance higher or equal to 5 Mb, and PIRs/DpnII fragments with lengths higher than or equal to 7500 bp.

## Bulk RNA-Seq and analysis

Total RNA from 16 AECs samples was sequenced in two batches (6 and 10 samples, respectively). In both cases, rRNA was depleted using the RiboCop Kit (Corall) and final libraries were prepared using the Total RNA-Seq Library Prep Kit (Corall). Samples were sequenced on NovaSeq6000 (NovaSeq Control Software 1.7.5/RTA v3.4.4) with a 51nt(Read1)-8nt(Index1)-8nt(Index2)-51nt(Read2) setup using 'NovaSeqXp' workflow in 'SP' mode flowcell. The Bcl to FastQ conversion was performed using bcl2fastq_v2.20.0.422 from the CASAVA software suite. The quality scale used is Sanger / phred33 / Illumina 1.8 + . FASTQ files were processed using the nf-core/rnaseq pipeline version 3.5 within the nf-core framework (Supplementary Table 3 and Supplementary Data 9)[41].

## ChIP-Seq and peak calling

Human ascending AEC from four patients were cross-linked with 1% formaldehyde for 10 min. Around 5 million cells were used for each experiment, and each experiment was performed in duplicates. All steps were performed at 4 °C unless otherwise indicated. Cells were

lysed in swelling buffers (100 mM Tris at pH 7.5, 10 mM KOAc, 15 mM MgOAc, 1% igepal, PIC) for 10 min followed by Dounce homogenization. Nuclei were pelleted at 3200 x *g* for 5 min and lysed in a modified RIPA buffer (PBS with 1% NP-40, 0.5% sodium deoxycholate, 0.1% SDS, 1 mM EDTA, PIC) for 10 min. Chromatin was sonicated to 150–300 bp using a Diagenode bioruptor. Insoluble material was removed by centrifugation at 34,000 x *g* for 10 min, followed by pre-cleaning with StaphA cells for 15 min. One microgram of primary antibody/5 million cells (Diagenode, C15410196-10) was incubated with precleared chromatin for 16 h. Anti-rabbit secondary antibodies (Millipore) were added for 1 h. Ten microliters of StaphA cells/5 million cells were added for 15 min at room temperature. StaphA cells were washed twice with dialysis buffer (50 mM Tris at pH 8, 2 mM EDTA, 0.2% sarkosyl) and four times with immunoprecipitation wash buffer (100 mM Tris at pH 8, 500 mM LiCl, 1% NP-40, 1% sodium deoxycholate). Chromatin was eluted off StaphA cells in 1% SDS and 50 mM NaHCO3. For re-ChIP, eluted chromatin was diluted 10-fold in modified RIPA without SDS, and a second primary antibody was added overnight. For ChIP-seq and ChIP-qPCR, 200 mM NaCl was added, and cross-links were reversed for 16 h at 67 °C. DNA was purified using a PCR purification kit (Qiagen), eluting in 50 μL of water. One microliter of ChIP DNA was used for qPCR. For ChIP-seq, libraries were prepared using the Mondrian (NuGen) and size-selected using Pippin Prep (Sage Science). Libraries were sequenced with 1 Å- 50 base pair read on the NextSeq 500 platform (Illumina Inc). FastQ files were then processed using the nf-core/chipseq/1.2.2 pipeline[41] (Supplementary Table 4 and Supplementary Data 3).

## Allele-specific analysis of the HiCap data

Variant calling from pre-made HiCap read alignments was performed using bcftools (v.1.21)[52] mpileup with --redo-BAQ --adjust-MQ 50 --gap-frac 0.05 --max-depth 10000 -A parameters to include read pairs that map far apart and bcftools call with --keep-alts --multiallelic-caller. The resulting sites were split into biallelic records using bcftools norm with --check-ref x -m - and filtered with bcftools filter -i "QUAL >= 10 & FORMAT/GQ >= 20 & FORMAT/DP >= 10" --SnpGap 3 --IndelGap 10, leaving only high-quality calls covered by 10 or more reads. Next, bcftools view -i 'GT ="alt"' --trim-alt-alleles followed by view -m2 -M2 -v snps was used to leave only biallelic SNVs annotated with bcftools +fill-tags -- -t all. Finally, filters for heterozygous variants located on the reference chromosomes with genotype quality ≥ 50, depth ≥10, and allelic counts ≥5 for each allele were applied to each individual separately with bcftools view -s <individual> -i 'FORMAT/AD[0:0] > 4 & FORMAT/AD[0:1] > 4 >="het"' -e 'GT = "mis" | FMT/DP < 10 | FMT/GQ < 50'. The allelic imbalance was assessed with MIXALIME (v.2.25.2)[28] using the beta negative binomial model, and individual-level adaptive coverage filters enabled by MIXALIME combine --adaptive-min-cover --group <individual>. Variants were then annotated with the significantly interacting enhancer and promoter regions derived from HiCap.

MIXALIME provides one-tailed *P*-values. As we were not focusing on the direction of the allelic imbalance, the minimal P-values (reflecting the imbalance towards either Ref or Alt allele) were doubled to mimic the significance of a two-tailed test. The gene-level *P*-values were obtained by combining *P*-values for all SNPs annotated with a particular gene using the logit method[53] implemented in the metap package (v.1.8). *P*-values of the single variant-per-gene were left as is. Next, the gene-level *P*-values were re-combined across individuals using the same logitp function from metap, followed by Benjamini−Hochberg FDR correction for the number of genes.

## Motif annotation of the single-nucleotide variants

522 PWMs of the highest A-B-C quality from the HOCOMOCO v12 non-redundant set of 523 PWMs[29] were used for motif annotation of the single-nucleotide variants. To this end, we employed PERFECTOS-APE v3.0.6[54] (https://github.com/autosome-ru/macro-perfectos-ape). Valid motif hits with the *P*-value ≤ 0.0005 (default threshold) were used to estimate the variant impact, the motif *P*-value log$_2$ fold change between the reference and the alternative alleles.

## scRNA-seq cell type enrichment *test 1* and *test 2*

We performed two statistical tests to assess the impact of regulatory mutations. Test 1 was applied to promoter mutations: using a two-tailed Fisher's exact test, we compared the proportion of concordant genes between the BAV and TAV cohorts. Test 2 focused on enhancer mutations: similarly, a two-tailed Fisher's exact test was used to evaluate the proportions of both concordantly and discordantly affected genes across the cohorts. In both tests, target genes and TFs were prefiltered based on their expression levels or differential expression status.

## Enformer

To estimate variant effects, we used the PyTorch implementation of Enformer[33] deep learning model by Phil Wang [https://github.com/lucidrains/enformer-pytorch] and the HF Accelerate framework [https://github.com/huggingface/accelerate]. We imported the model weights from https://huggingface.co/EleutherAI/enformer-official-rough. Each variant was centered within the 196,608-nucleotide input window of the Enformer model and embedded in a genomic context of the same size. Only four central bins out of the 896 output prediction bins (each comprising 128 nucleotides) were retained, resulting in a prediction window of 512 nucleotides. Enformer predictions were restricted to human-specific genome tracks. Each variant was scored on both the forward and reverse complement strands. The described procedure was performed for both the reference and alternative alleles. The predicted effect of a variant was computed as the difference between the log-transformed mean of the Alt allele bins and the log-transformed mean of the Ref allele bins (using a $log(x + 1)$ transformation). The means were calculated after aggregating data across all relevant bins for each allele. Next, we performed Z-score transformation of these values using a background set of 396,440 frequent single-nucleotide variants (MAF > 0.01 in at least one population, GNOMAD v4.1.0) from chromosome 22, which are largely considered to comprise a neutral set. In the end, we used four subsets of epigenomic features for each variant: (1) DNase, ChIP (CTCF, H3K4me1, H3K4me3, H3K9me3, H3K27me3, H3K27ac, H3K36me3) for ascending aorta and CAGE for aortic endothelial cells, 15 × 2 features in total; (2) DNase, ChIP (CTCF, H3K4me1, H3K4me3, H3K9me3, H3K27me3, H3K36me3) and CAGE for K562, amounting to 22 × 2 features; (3) DNase, ChIP (H3K4me1, H3K4me3, H3K9me3, H3K27me3, H3K36me3) and CAGE for adult hearts and heart left ventricles, 13 × 2 features in total; (4) DNase, ChIP (H3K4me1, H3K4me3, H3K9me3, H3K27me3, H3K36me3) and CAGE for fetal/embryo hearts, 23 × 2 features in total;

## Protein-protein interaction network construction, visualization and GO terms enrichments

BAV and TAV specific protein-protein interaction networks were generated using STRINGdb v12[48]. For each network, BAV- and TAV-specific genes affected by regulatory mutations were used as input. The following settings were applied: full STRING network, active interaction sources - text mining, experiments and databases, confidence threshold - 0.400 and one shell of interactors with a maximum of one interactor. Disconnected nodes were removed to improve interpretability. Resulting networks were visualized using Cytoscape v3.10.3[55]. GO terms enrichments were obtained from STRINGdb using the relevant background genes.

## Ethics

The study was approved by the Human Research Ethics Committee at Karolinska Institutet (application number 2006/784−31/1 and 2012/1633−31/4), Stockholm, Sweden; written informed consent was obtained from all the individuals according to the Declaration of

Helsinki. All the methods were carried out following relevant guidelines, including the genetic studies, complying with international rules on genetics 2.

## Reporting summary

Further information on research design is available in the Nature Portfolio Reporting Summary linked to this article.

## Data availability

Raw DNA and RNA sequencing read files generated in this study are regulated by the Swedish Law on Patient Data (2008:355). They are available under controlled access because they are derived from human biological samples and contain individual-level genomic sequence information that may enable participant re-identification. Participant consent permits data sharing only via controlled-access procedures. Access may be granted to qualified researchers for research purposes. Requests should include a brief research proposal, intended data use, and confirmation of appropriate ethical approvals. Requests should be directed to the corresponding author, hanna.bjorck@ki.se. Requests will be acknowledged within 5 working days and are typically reviewed within 30 days. Approved applicants will be required to sign a data use agreement restricting use of the data to the approved project and prohibiting attempts to re-identify participants or redistribute the data to third parties. Intermediate, anonymized data required to replicate the analysis are provided in Supplementary Tables and Supplementary Data. Source data are provided in this paper.

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

## Acknowledgements

The authors acknowledge support from the National Genomics Infrastructure in Stockholm, funded by Science for Life Laboratory, the Knut and Alice Wallenberg Foundation, and the Swedish Research Council, and SNIC/Uppsala Multidisciplinary Center for Advanced Computational Science for assistance with massively parallel sequencing and access to the UPPMAX computational infrastructure. P.S. was supported by the European Union's Horizon 2020 Research and Innovation Program under the Marie Sklodowska-Curie Grant Agreement No 860002. H.B. was supported by the Swedish Heart Lung Foundation grants #20240686, #20240685, and #20240180, and the donation from Mr. Fredrik Lundberg. The information contained in this study reflects only the authors' views. The European Commission and the European Research Executive Agency do not accept responsibility for any use made of the information contained therein.

## Author contributions

Conceptualization: P.S. and A.Z. Methodology: P.S., A.Z., I.V.K., and E.L. Software: A.Z., N.G., A.B., V.N., M.P.S., and S.P. Validation: E.L., R.M., and D.S. Formal analysis: A.Z., N.G., A.B., V.N., S.P., and M.P.S. Investigation: A.Z., R.S., and K.L. Resources: A.F-C., P.S., P.E., H.M.B., and J.L. Data curation: A.Z. and P.S. Writing – original draft: A.Z. Writing – review and editing: A.Z., P.S., H.M.B., P.E., I.V.K., E.L., M.P.S., N.G., A.B., V.N., R.M., and K.L. Visualization: A.Z., M. P. S., R.M., A.B., and V.N. Supervision: P.S., I.V.K., H.M.B., and P.E. Project administration: A.Z. and P.S. Funding acquisition: P.S. and H.M.B.

## Funding

## Competing interests

P.S. holds the active patent for targeted chromosome information detection used as a part of HiCap (EP2984182A1; Inventors: Rickard Sandberg, Pelin Sahlén). All other authors declare that they have no competing interests.
