## [Transparent Peer Review file · Nature Communications]

Rare regulatory mutations disrupt mesenchymal molecular programs driving endocardial cushion formation in bicuspid aortic valve

Corresponding Author: Dr Pelin Sahlén

Version 0:

Reviewer comments:

Reviewer #1

(Remarks to the Author)

Zhigulev et al. present a WGS analysis of 8 BAV cases through the lens of joint transcriptomic and epigenetic analyses. They identify enrichment of a gene network that is perturbed by promoter and enhancer variants and may affect aortic valve development. However, their results are undermined by several significant limitations.

In particular, 3 of the 8 included BAV patients have significant genetic lesions that make them unrepresentative of the broader BAV population. That raises serious questions about the generalizability of their conclusions.

1. In Figure S6, a total of 8/16 patients were removed as outliers after MDS analysis of RNAseq data. In fact, 2/8 BAV patients clustered with TAV patients in the initial MDS plot, and these relationships were discarded prior to the analysis. This extensive data curation could introduce bias into the downstream analyses and call into question the internal and external validity. Were the trends observed in the DGE data similar in the excluded samples? Why or why not?

2. The tier 1 and tier 2 gene lists used to curate BAV candidate genes are confusing. For example, the tier 1 gene list is derived from a disease ontology term that contains only 4 genes with rare variants that have been implicated in human BAV disease and a GWAS catalog that contains 2 genes with common variant associations. These common variants are not relevant to the types of variants that are queried in this study. Additionally, genes cited as "tier 1" such as EDNRA, PUF60, SNIPI, SNIP1, and SMC3 have little genetic evidence linking them to human BAV disease. Please repeat the enrichments using tier 1 genes that are derived from recurrent observations in human data. For all analyses, please also create random gene lists of equal length and repeat enrichment tests in BAV and TAV cases.

3. HiCAP and CHIP-seq were performed on primary ECs explanted from aortic tissue and expanded in culture. How faithfully do the promoter-enhancer interactions of isolated ECs cultured in a static microenvironment recapitulate in vivo mechanisms? Does culture expansion cause significant alterations in promoter activity, and would such changes be predictable compared to the in vivo states of the ECs? How many promoter-enhancer observations may be concordant due to convergent effects of cell manipulation?

4. Throughout this report, the authors refer to promoter and enhancer variants as "causative mutations." The evidence that they present is insufficient to determine causality or pathogenicity. They describe genetic modifiers that perturb a regulatory network of genes that promotes aortic valve development. Similarly, their results do not directly address the "missing heritability" of BAV disease and this should not be stated definitively. Please correct these statements, revise the title, and provide the appropriate perspective on these findings in a discussion of the limitations.

5. Concordance and discordance of promoter variants and enhancer interactions was classified based on only 16 observations with a MAF cutoff of 10%. Therefore, many observations would be expected to occur by chance. How did the authors account for this?

6. Table S1 indicates that one of the TAV participants has Marfan syndrome. FBN1 mutations perturb transcription of aortic ECs and SMCs due to impairment of cell interactions with the ECM and would therefore be expected to cause global alterations of gene expression. In Table S2, one participant (BAV7) is listed as having structural variants that affect 6 different genes on multiple chromosomes. Another participant (BAV5) has a frameshift mutation in BRAF, which causes CFC syndrome. Why weren't these individuals excluded? Did all participants undergo a genetic evaluation to exclude syndromic forms of aortic disease?

7. The authors state that they predicted the transcriptomic impact of altered promoter-enhancer interactions using a developmental RNAseq dataset. They assert that "Art_EC_1, the endothelial cell state with the closest transcriptomic profile to adult endothelial cells, showed greater similarity to mesenchymal cells and fibroblasts than to other endothelial populations." How did they classify which transcripts and cell types were affected, and how confident are they in the equivalence of cell populations between adult and embryonic cell states? Please specify in greater detail.

Reviewer #2

(Remarks to the Author)

The authors sought to assess genetics causes of bicuspid aortic valve based on a first integrative whole-genome interactome analysis of BAV and TAV patients.

- Methodology: collaborative study, based on human tissue and complex cell and genetic approach
- Ethics committee ok, written informed consent ok, consent for genetics ?
- 8 BAV patients + 8 TAV patients undergoing open heart surgery with aorta biopsy
- Primary endpoint: to identify causative regulatory interactions and address the missing heritability of BAV
- Results: genes implicated in aortic valve development are enriched for gain-of-function regulatory mutations in BAV patients, rare regulatory mutations in BAV patients alter the transcriptome of fetal heart mesenchymal cells and fibroblasts
- Conclusion: critical role of rare regulatory mutations in BAV pathology which influence developmental mesenchymal and fibroblasts cells involved in aortic valve formation

I congratulate authors for this nice methodology and manuscript. I have nevertheless a few comments or questions:

1) Abstract, 2nd sentence, and introduction line 39: based on an old publication including child with BAV and several cardiac malformations, you state that heritability of BAV is very high. However, other publications with less selected patients report a lower prevalence of BAV in relatives with a lower heritability (for example Galian-Gay L, Heart 2019, Boureau AS, Int J Cardiol 2022). Please make appropriate correction to your manuscript.

2) Please provide more information about the selected patients: what were the indications for the operation? Were the patients operated on because of valvular or aortic disease? These patients seem very selected (aortic dilatation), they cannot represent the full spectrum of BAV pathology. Please comment and add a limitation section in your manuscript.

3) Although data are promising, the number of patients is limited. Do you think you can extrapolate your findings to all patients with BAV or only to a specific subpopulation (those with aorta dilatation for example) ? Please comment and expand the limitation section in your manuscript.

4) Have the patients given their consent for the genetic studies and does the study comply with international rules on genetics? Please add this information to your manuscript

5) Beyond the pathophysiological aspect, do you think that your results could have, in one way or another, a significant impact on clinical practice: patient categorization, risk scoring, familial screening, etc...? Please elaborate.

Reviewer #3

(Remarks to the Author)

Please find the report uploaded as a pdf file.

Reviewer #4

(Remarks to the Author)

This manuscript presents an integrative genomic study aimed at uncovering the contribution of rare non-coding regulatory mutations to the heritability of BAV, a common congenital heart defect. By combining promoter capture Hi-C with whole-genome sequencing from patient-derived aortic endothelial cells, and integrating these data with developmental single-cell and spatial transcriptomics, the authors uncover gain-of-function regulatory interactions that implicate novel genes and developmental cell types in BAV pathogenesis.

Major Comments

1. Significance and Novelty

This study addresses an important and underexplored source of 'missing heritability' in BAV by focusing on rare regulatory

variants. The integration of patient-specific chromatin interaction maps with WGS data is a notable strength and a methodological advance over proximity-based mapping typically used in GWAS. Additionally, the linkage to relevant fetal cell states through developmental transcriptomic data strengthens the mechanistic interpretation.

2. Data Support and Interpretation

The findings are generally well-supported by data, but the functional consequences of the regulatory mutations remain largely inferential. Although TF binding motif disruption was evaluated, only two cases involved disruption of known binding sites. Most cases are labeled as gain-of-function based on the presence of new interactions, which could benefit from experimental validation or orthogonal functional evidence.

3. Cohort Size and Generalizability

The cohort consists of 8 BAV and 8 TAV individuals. While the depth of multi-omic profiling is impressive, the small sample size may limit generalizability. The authors should more explicitly acknowledge this limitation and discuss the potential for replication in larger or independent cohorts.

4. Causal Inference Criteria

The strategy to define 'concordant' mutations and label them as causative is reasonable but may benefit from additional detail or refinement.

5. Methodological Strengths

The manuscript is very detailed in methodology and reproducibility, integrating high-quality pipelines and developmentally relevant transcriptomic data.

Minor Comments

- The terms 'concordant' and 'discordant' should be defined more clearly in the main text.
- Consider including locus-specific examples of newly formed regulatory interactions.
- Provide a summary table of top BAV-specific regulatory variants.
- Emphasize limitations of using adult endothelial cells as proxies for fetal activity.

Recommendation

Minor Revision

This is a strong and timely study with high relevance to cardiovascular genetics and regulatory genomics. The manuscript is methodologically sound, well-written, and presents novel biological insights. Pending minor revisions and clarifications, this work merits publication.

Additional Suggestion for Enhancement

While the integration of HiCap and WGS in this study provides valuable mechanistic insight into promoter-enhancer interactions in BAV, the authors may consider complementing this approach with sequence-based functional prediction of promoter variants. In particular, the recently developed PromoterAI model (Jaganathan et al., Science, 2025) offers robust prediction of gene expression changes from non-coding promoter variants using only DNA sequence. Such complementary analysis could strengthen causal inference and broaden discovery potential, particularly given the current limitations in sample size and interaction resolution.

Figure Reference Clarification

There appears to be a labeling error in the reference to "Fig. 3h" describing transcription factor (TF) binding motif disruption. Based on context, this likely refers to a schematic depicted in Fig. 2h. However, no panel labeled '2h' is actually present in the manuscript figures, despite this citation appearing in both the text and the caption for Figure 2. It would benefit the clarity and accuracy of the manuscript to revise this reference and ensure that all figure panels are correctly labeled and accounted for.

Supplemental Materials and File Organization

In addition, many of the supplemental figures and files are very large in size. For example, Supplemental Figure 5 exceeds 1 GB, and I was unable to download or view it. The supplemental materials overall would benefit from streamlining and cleanup. There are numerous supplemental tables, but a clear and centralized description of their contents and organization is absent as far as I could tell.

Version 1:

Reviewer comments:

Reviewer #1

(Remarks to the Author)

I read through the response to the reviewers and agree that the authors satisfactorily addressed all my comments except for one. Please exclude the one TAV sample with an FBN1 PV, one BAV sample with a frameshift PV in BRAF and one BAV sample with multiple genomic structural variants. The problem with including these samples in the analysis is not, as the authors asserted, that they are not representative of BAV cases in general, but that the genetic lesions cause non-BAV

developmental abnormalities that will confound their analysis.

Reviewer #2

(Remarks to the Author)

The authors performed appropriate modifications to their manuscript. I have no further comments.

Reviewer #3

(Remarks to the Author)

The authors have fully addressed my concerns.

One formatting issue is that the text is obscured and difficult to read in the new network figure, Fig. 4c. Please fix this before publication.

Reviewer #4

(Remarks to the Author)

Dear Authors,

Thank you for your thoughtful and thorough responses to the reviewer comments, and for the substantial revisions made to the manuscript.

I appreciated the clarity and transparency in the way you addressed the concerns raised. The revisions to terminology, particularly moving away from “causative” and “gain-of-function” language, improve the scientific precision of the manuscript. I also thought the reanalysis of the RNA-seq data, refinement of the tiered gene lists, and the expanded methodological detail all contributed to a stronger and more reproducible study.

The use of Enformer as an orthogonal tool to support regulatory variant impact is a smart addition, and your expanded discussion on limitations - especially regarding the use of cultured adult endothelial cells - was well stated. I also appreciated the fixes to figure labeling, the clarification of “concordant” vs. “discordant” events, and the streamlining of supplemental materials.

While some interpretive gaps remain (e.g., inclusion of potentially syndromic individuals), I think you've handled them reasonably and with scientific caution.

Overall, the manuscript has improved substantially, and I support its publication.

Version 2:

Reviewer comments:

Reviewer #1

(Remarks to the Author)

I am satisfied that the authors addressed my remaining critique. Congratulations on this important work!

Rare regulatory mutations disrupt mesenchymal molecular programs driving endocardial cushion formation in bicuspid aortic valve patients

Response to Referees Letter

We are pleased to resubmit our revised manuscript (ID NCOMMS-25-17709-T), titled “Rare regulatory mutations disrupt mesenchymal molecular programs driving endocardial cushion formation in bicuspid aortic valve patients”, to *Nature Communications*. We sincerely thank the reviewers and the editorial team for their thorough evaluation and constructive feedback on our initial submission. Below, we respond point-by-point to the reviewers' comments. Our responses are marked in blue, and specific sections where revisions were made are indicated for clarity.

Reviewer #1 (Remarks to the Author):

Zhigulev et al. present a WGS analysis of 8 BAV cases through the lens of joint transcriptomic and epigenetic analyses. They identify enrichment of a gene network that is perturbed by promoter and enhancer variants and may affect aortic valve development. However, their results are undermined by several significant limitations.

In particular, 3 of the 8 included BAV patients have significant genetic lesions that make them unrepresentative of the broader BAV population. That raises serious questions about the generalizability of their conclusions.

1. In Figure S6, a total of 8/16 patients were removed as outliers after MDS analysis of RNAseq data. In fact, 2/8 BAV patients clustered with TAV patients in the initial MDS plot, and these relationships were discarded prior to the analysis. This extensive data curation could introduce bias into the downstream analyses and call into question the internal and external validity. Were the trends observed in the DGE data similar in the excluded samples? Why or why not?

We thank the reviewer for this important observation regarding our bulk RNA-seq dataset. As noted in the Methods section, a substantial difference in sequencing depth between the old and new batches resulted in several samples being flagged as outliers during MDS analysis.

Specifically, samples from the older batch exhibited a high proportion of genes with zero counts, making the batch correction infeasible. For this reason, these samples were formerly excluded from further quantitative analyses.

We want to clarify that the bulk RNA-seq dataset was **not used for any core analyses** or direct comparisons between the BAV and TAV cohorts. Accordingly, the main manuscript did not include a dedicated Results section for this dataset.

Instead, the bulk RNA-seq data served solely as a **supporting internal control**, used in a limited manner in two figures (Fig. 2a-b and Fig. 3a), where aggregated gene counts from patients with high-quality transcriptomic profiles were compared with other datasets. As shown in the updated figures (Fig. 2a-b and Fig. 3c, respectively), the trends remained stable regardless of whether all samples or only the high-quality subset were used, indicating that these aggregated results were **unaffected by outlier exclusion**.

Our original intent was to maintain transparency by including basic exploratory analysis, such as MDS and DGE, in the Supplementary Materials. However, since multiple reviewers raised concerns about potential confusion, we have removed this analysis from the manuscript entirely. We now use **aggregated gene counts from all 16 patients** in Figures 2a-b and 3c.

2. The tier 1 and tier 2 gene lists used to curate BAV candidate genes are confusing. For example, the tier 1 gene list is derived from a disease ontology term that contains only 4 genes with rare variants that have been implicated in human BAV disease and a GWAS catalog that contains 2 genes with common variant associations. These common variants are not relevant to the types of variants that are queried in this study. Additionally, genes cited as "tier 1" such as EDNRA, PUF60, SNIPI, SNIP1, and SMC3 have little genetic evidence linking them to human BAV disease. Please repeat the enrichments using tier 1 genes that are derived from recurrent observations in human data. For all analyses, please also create random gene lists of equal length and repeat enrichment tests in BAV and TAV cases.

We thank the reviewer for carefully examining our *tier 1* and *tier 2* gene lists. Our original intention in using predefined public gene sets was to minimize bias in candidate gene selection and enable a broadly inclusive analysis of potential contributors to BAV pathology. However, we recognize

that including genes with limited or indirect relevance to human BAV could weaken the interpretability of the enrichment results and potentially cause confusion for readers.

In response to the reviewer's feedback, we have **revised the tier 1 gene list** to include only genes with recurrent and well-established associations with BAV in human genetic studies. The revised *tier 1* set now includes: **GATA4, NOTCH1, GATA6, ROBO4, SMAD6, ADAMTS19, and TBX5**. *Tier 2* genes now represent BAV-related or general aortic valve subsets (**HP:0001647, DOID:0080332, GO:0003176, HP:0001646**). We have re-run all enrichment analyses using the updated *tier 1* gene set and, where appropriate, used a combined *tier1+tier2* set for statistical purposes.

The reviewer also highlighted the importance of using a random gene list of equal length as a control for the enrichment analysis. We used **all genes detected in the relevant experiment as the background set for statistical comparison**. We have added a corresponding clarification to the Methods section ("Protein-Protein Interaction Network Construction, Visualization, and GO Term Enrichment").

3. HiCAP and CHIP-seq were performed on primary ECs explanted from aortic tissue and expanded in culture. How faithfully do the promoter-enhancer interactions of isolated ECs cultured in a static microenvironment recapitulate *in vivo* mechanisms? Does culture expansion cause significant alterations in promoter activity, and would such changes be predictable compared to the *in vivo* states of the ECs? How many promoter-enhancer observations may be concordant due to convergent effects of cell manipulation?

We thank the reviewer for this important and thoughtful question regarding the biological fidelity of our *in vitro* endothelial cell model. We fully acknowledge that endothelial cells cultivated *in vitro* under static conditions may not fully reproduce the mechanical and microenvironmental context of the *in vivo* aortic endothelium. However, culture expansion was technically necessary due to the high input requirement of the HiCap protocol, which requires approximately one million cells per sample.

To support the physiological relevance of our dataset, we show in Fig. 2a-b that promoter activity patterns of endothelial-specific genes depend on their interaction status, demonstrating that our ECs retain essential endothelial characteristics. Furthermore, Fig. 2c shows a significant overlap

between enhancer profiles of patient-derived ECs and the endothelial model cell line TeloHAEC. While these trends are already strong, we further increase biological relevance in downstream analyses through additional filtering steps, including TF binding and expression integration.

Importantly, all cultures were handled uniformly, ensuring that any potential culture-induced effects would be consistent across both BAV and TAV cohorts and therefore unlikely to bias comparisons between them. If such effects were present, both concordant and discordant promoter–enhancer pairs would be expected to be affected similarly. Instead, we observed a specific enrichment of concordant cases in BAV samples, suggesting that the detected signal reflects biologically meaningful differences rather than culture-related artifacts.

We have added this clarification to the revised Discussion section (line 378), where we now explicitly describe the technical limitations associated with cell culturing, the measures taken to preserve biological fidelity, and observations resulting from culture expansion.

4. Throughout this report, the authors refer to promoter and enhancer variants as "causative mutations." The evidence that they present is insufficient to determine causality or pathogenicity. They describe genetic modifiers that perturb a regulatory network of genes that promotes aortic valve development. Similarly, their results do not directly address the "missing heritability" of BAV disease and this should not be stated definitively. Please correct these statements, revise the title, and provide the appropriate perspective on these findings in a discussion of the limitations.

We thank the reviewer for this thoughtful comment. In response, we **revised the title** and the terminology throughout the manuscript. We now use the term **“non-coding variant”** to describe general genetic variation, and **“regulatory mutation”** when functional evidence supports an effect on gene regulation. Additionally, we have expanded the Discussion section to provide an appropriate perspective on these findings, emphasizing their contribution to understanding regulatory networks without making definitive claims about causality or missing heritability.

5. Concordance and discordance of promoter variants and enhancer interactions was classified based on only 16 observations with a MAF cutoff of 10%. Therefore, many observations would be expected to occur by chance. How did the authors account for this?

We thank the reviewer for this important observation. We agree that, at the initial stage of concordant and discordant subset generation, based on the overlap of HiCap and WGS data, some observations may occur by chance, particularly given the modest sample size (n=16) and the relatively high MAF cutoff of 10%.

However, we took several steps to mitigate the impact of chance associations and strengthen the biological relevance of our findings:

Initial signal enrichment:

As shown in Table 1, the BAV cohort exhibited a modestly higher number of promoter–enhancer interactions than TAV and an equal number of non-coding variants. Notably, only concordant cases showed enrichment in the BAV cohort. In contrast, discordant cases did not, suggesting we detect a non-random, biologically meaningful signal even at this early stage.

Filtering by functional criteria:

We refined these initial cases by integrating transcription factor (TF) binding affinity predictions and expression data from developmental scRNA-seq, requiring the affected TF and the target gene to be expressed in the relevant cell types or states. This filtering substantially reduces the likelihood of false positives due to spurious variant–gene pairings.

Allele frequency considerations:

Although we initially used a MAF cutoff of 10%, the strongest signal ultimately emerged within variants with **MAF < 2% for promoters** and **MAF < 4% for enhancers**, resulting in a much smaller number of observed cases.

Use of internal controls:

To further validate the robustness of our findings, we employed multiple internal controls, including:

- BAV vs. TAV comparisons
- Concordant vs. discordant interactions

- Rare vs. less-rare mutations
- Cross-validation across both developmental scRNA-seq datasets (blood-derived endothelial populations) and adult bulk RNA-seq data from ascending aorta tissues (Figure 3C)

These analyses consistently showed that concordant cases were preferentially enriched in BAV and linked to mesenchymal transcriptomic profiles.

These multi-layered filtering and control strategies substantially reduce the likelihood that our results are driven by random chance. We have clarified these points throughout the revised manuscript.

6. Table S1 indicates that one of the TAV participants has Marfan syndrome. FBN1 mutations perturb transcription of aortic ECs and SMCs due to impairment of cell interactions with the ECM and would therefore be expected to cause global alterations of gene expression. In Table S2, one participant (BAV7) is listed as having structural variants that affect 6 different genes on multiple chromosomes. Another participant (BAV5) has a frameshift mutation in BRAF, which causes CFC syndrome. Why weren't these individuals excluded? Did all participants undergo a genetic evaluation to exclude syndromic forms of aortic disease?

We thank the reviewer for raising this critical point. Patients were not excluded because they did not exhibit an obvious syndromic form of the disease beyond having BAV. Given that most BAV cases are likely to have a genetic component, it would not be feasible to exclude these patients.

We acknowledge that our findings **cannot explain the full genetic spectrum of BAV**, instead, they provide insight into a subset of regulatory changes contributing to the disease. This limitation is now explicitly stated in the revised Discussion section (line 372).

7. The authors state that they predicted the transcriptomic impact of altered promoter-enhancer interactions using a developmental RNAseq dataset. They assert that "Art_EC_1, the endothelial cell state with the closest transcriptomic profile to adult endothelial cells, showed greater similarity to mesenchymal cells and fibroblasts than to other endothelial populations." How did they classify which transcripts and cell types were affected, and how confident are they in the equivalence of cell populations between adult and embryonic cell states? Please specify in greater detail.

Once again, we thank the reviewer for this very relevant and thoughtful comment. As noted by all reviewers, our manuscript employs several nontraditional approaches to data integration, and we agree that a more precise description of our statistical methods strengthens the manuscript.

To classify a transcript as "**affected**" by altered promoter–enhancer interactions, we applied the following criteria:

1. The transcript is involved in a **concordant event**;
2. The concordant interaction is active in a given cellular context:
 - The underlying regulatory variant significantly alters TF binding affinity;
 - The target transcript and the affected TF are expressed in the specific tested cell type or state.

We then quantified the number of affected transcripts in each cell type for both BAV and TAV cohorts, and for concordant versus discordant events, and applied a two-tailed Fisher's exact test to assess enrichment. FDR correction was used for enhancer-driven concordant cases to account for multiple hypothesis testing. Raw p-values were reported for promoter-associated cases due to the limited number of observations, which precluded reliable adjustment. We used DEGs for filtering to emphasize transcriptomic specificity across cell types and states. This approach is now clarified in the main text (Figure 3A).

We acknowledge the reviewer's concern regarding the equivalence of embryonic and adult endothelial populations. The developmental connection was drawn based on **consistent localization of these cells in the vascular tree**, specifically the internal layer of the outflow tract and great arteries in the developmental dataset, and the aortic endothelium as the collection site for adult cells. This endothelial population differentiates during the first trimester of intrauterine development, without any known major diversification events later, supporting the biological validity of this comparison. To clarify this point, we have added **Supplementary Fig. S2** and highlighted it in both the Results and Discussion sections.

Reviewer #2 (Remarks to the Author):

The authors sought to assess genetics causes of bicuspid aortic valve based on a first integrative whole-genome interactome analysis of BAV and TAV patients.

- Methodology: collaborative study, based on human tissue and complex cell and genetic approach
- Ethics committee ok, written informed consent ok, consent for genetics ?
- 8 BAV patients + 8 TAV patients undergoing open heart surgery with aorta biopsy
- Primary endpoint: to identify causative regulatory interactions and address the missing heritability of BAV
- Results: genes implicated in aortic valve development are enriched for gain-of-function regulatory mutations in BAV patients, rare regulatory mutations in BAV patients alter the transcriptome of fetal heart mesenchymal cells and fibroblasts
- Conclusion: critical role of rare regulatory mutations in BAV pathology which influence developmental mesenchymal and fibroblasts cells involved in aortic valve formation

I congratulate authors for this nice methodology and manuscript. I have nevertheless a few comments or questions:

1) Abstract, 2nd sentence, and introduction line 39: based on an old publication including child with BAV and several cardiac malformations, you state that heritability of BAV is very high. However, other publications with less selected of BAV in relatives with a lower heritability (for example Galian-Gay L, Heart 2019, Boureau AS, Int J Cardiol 2022). Please make appropriate correction to your manuscript.

We thank the reviewer for this comment. The prevalence of BAV in first-degree relatives has been reported to be up to 7% (PMID: 37288540) and higher in families (up to 23%). While the heritability of BAV is described as high in Galian-Gay L 2019 and other studies, we have **toned down the statement** in our manuscript, changing “high” to “**substantial**” prevalence to reflect a more balanced view.

2) Please provide more information about the selected patients: what were the indications for the operation? Were the patients operated on because of valvular or aortic disease? These patients seem very selected (aortic dilatation), they cannot represent the full spectrum of BAV pathology. Please comment and add a limitation section in your manuscript.

Patients included in this study all underwent **ascending aortic surgery**, with or without concomitant aortic valve replacement. Those operated exclusively for aortic valve disease were not included, due to the difficulty of isolating endothelial cells from the limited size of non-dilated aortic biopsies.

Hence, we fully agree with the reviewer that our results may **not be representative of the entire BAV population** but provide insight into BAV formation in individuals with concomitant aortic dilatation. The Discussion section (line 372) has been updated to address this limitation, and the Supplementary Table 1 has been expanded accordingly.

3) Although data are promising, the number of patients is limited. Do you think you can extrapolate your findings to all patients with BAV or only to a specific subpopulation (those with aorta dilatation for example) ? Please comment and expand the limitation section in your manuscript.

Please see our response to Comment 2.

4) Have the patients given their consent for the genetic studies and does the study comply with international rules on genetics? Please add this information to your manuscript.

We thank the reviewer for raising this important point. Yes, patients provided all relevant consents, including for genetic studies. All genetic analyses comply with international rules and relevant guidelines. This information has been added to the Methods section under “Patient cohorts.”

5) Beyond the pathophysiological aspect, do you think that your results could have, in one way or another, a significant impact on clinical practice: patient categorization, risk scoring, familial screening, etc...? Please elaborate.

We thank the reviewer for this insightful comment. This work advances the understanding of BAV formation and BAV-associated aortopathy, supporting the view that BAV has a **substantial genetic component**. However, the genetic architecture is complex and heterogeneous. These findings may have implications for **future screening strategies**, including the identification of novel targets for familial genetic testing. The Discussion section (line 399) has been updated accordingly.

Reviewer #3 (Remarks to the Author):

Key results

The authors investigate biopsies taken from 8 patients with bicuspid aortic valve (BAV) vs 8 patients with tricuspid aortic valve (TAV), by performing whole genome sequencing as well as promoter capture experiments (using the HiCap methodology) and analysing and integrating these data with single-cell RNA-seq and spatial transcriptomics data. They identify mutations (from WGS data) that are concordant with allele specific interactions (from HiCap data) at the variant and gene levels. They then find that concordant genes with rare mutations (MAF<3%) show differential expression in relevant cell populations in single-cell RNA-seq data and spatial transcriptomics data, suggesting a role for these putative regulatory rare mutations. They then perform pathway analysis and find that genes with concordantly gained mutation and interaction are involved in relevant pathways for endothelial cell development (such as TGFB and EMT). The genes affected vary across patients, which suggests a heterogeneity of potential genetic markers for the disease.

Significance

The study design of combining whole genome sequencing and chromosome conformation capture data to identify variants with (putative) regulatory interactions is a very valuable and relevant task with great potential for identifying new putative genetic risk factors. I also thought that it was very interesting that these putative regulatory interactions were linked to genes differentially expressed by mesenchymal/fibroblast cell types around the aortic valve in the single-cell RNA-seq and spatial transcriptomics data.

However, the title suggests that “rare gain-of-function regulatory mutations explain the missing heritability of bicuspid aortic valve”, but in my view, the manuscript falls short of this promise, for the following reasons:

1. There is some evidence but no proof that the interactions that are concordant with the identified mutations are regulatory interactions. The authors show that they have increasing overlaps with H3K27ac marks. However, there is no single histone modification mark that is exclusively

responsible for enhancer activity, see, for example, in Pollex and Furlong, Mol. Cell. 66(4):439-441 (2017), and functional studies would be needed to prove causation.

We thank the reviewer for this meaningful and insightful comment. We agree that, while **H3K27ac** is a well-established marker of active enhancers, it does not exclusively indicate enhancer function. Our study used the overlap with H3K27ac primarily to **validate the methodology**, rather than as evidence that the identified interactions are causative developmental enhancers. We also note that such pathological developmental interactions would unlikely show strong regulatory signals in adult tissue.

Nonetheless, our **multi-step prioritization strategy** strengthens the case that these interactions are regulatory. This includes:

- Concordance with variants combined with identification of potentially active events (TF affinity changes and expression of both the target gene and affected TF in developmental scRNA-seq data);
- A multi-layered control setup (BAV vs. TAV, concordant vs. discordant events, common vs. rare variants, and aortic tissue vs. blood-derived transcriptomes);
- And the addition of ***in silico* functional validation using the deep learning model Enformer (line 209)**.

Together, these approaches provide strong support for considering the interactions as regulatory. At the same time, we recognize the **limitations of these inferences**, and the Discussion section (line 355) has been updated to communicate this clearly to readers.

2. I am not convinced that the presented analysis has sufficient evidence that these mutations are gain- of-function. Regulatory interactions may lead to gain-of-function or loss-of-function of the affected gene, and I could not see a consideration of this.

We thank the reviewer for this crucial comment. To avoid confusion, we have **replaced the terms “gain-of-function” (GoF) and “loss-of-function” (LoF) with “gain-of-interaction” and “loss-of-interaction”**, respectively. Fig. 2f now illustrates these cases, and the manuscript clearly discusses this terminology and its interpretation (line 137).

3. While the identified concordant rare mutations may explain some of the missing heritability, I did not see any evidence that the identified concordant rare mutations with from 8 BAV patients vs 8 TAV controls explain “the” (i.e. all) missing heritability of the BAV phenotype.

We agree with this comment. Accordingly, we have **toned down the conclusions about heritability throughout the manuscript** and explicitly discuss the **limitations of our cohort** in the Discussion section (line 372), clarifying that our findings provide insight into a subset of regulatory changes rather than the complete genetic architecture of BAV.

Validity

I have the impression that **gain-of-function** is used interchangeably with gain-of-interaction, which is very different and I find this a source of major concern for the analytical approach. Gained regulatory interactions may be enhancing gene expression (gain-of-function), repressing gene expression (loss-of-function) or not have a quantitative effect.

4. For the definition of **concordant interactions**, I found contradicting information in the manuscript. Fig. 2f suggests that only interactions gained for the mutation (minor allele) are taken into consideration, while the text suggests that those were considered where “the variant affected the interaction”. For a functional understanding of these changing interactions, it would be important to consider both types of **rewiring of contacts**: gained and lost interactions separately.

We thank the reviewer for this comment. As clarified in our response to Comment 2, we now consistently use the terminology “**gain-of-interaction**” and “**loss-of-interaction**” to capture both types of rewiring separately. Fig. 2f has been updated to illustrate these cases. We also note that **gain-of-interaction does not directly equate to gain-of-function or loss-of-function**, and this distinction is now clearly stated in the manuscript.

5. It would also be important to also analyse these rewired interactions for chromatin state (enhancing vs repressing). Simply looking at gained interactions would average over the gain-of-function and loss-of-function effects, reducing the signal.

We thank the reviewer for this comment. The **chromatin states of rewired interactions in adult cells may not correspond to their states during valve formation**. However, as mentioned in the

Discussion section (line 387), approaches such as **Motif Activity Response Analysis** could help infer the activating or repressing nature of bivalent TFs, thereby annotating regulatory regions *in silico* as enhancing or repressing. Implementing this approach would require **advanced datasets capable of quantifying patient-specific promoter activity and optimizing enhancer analysis**, representing a separate research question for future studies.

6. There is another scenario which is not considered by the manuscript, in which one allele is mutated and there is no rewiring of contacts (i.e. the interaction is maintained) but there is a change in chromatin state of the regulatory region (***recolouring of chromatin states***) as a consequence of the mutation. Considering these might further help identify previously unknown mutations that have a role in regulatory interactions.

An example study that considers the effects of chromatin rewiring and recolouring (in a different context) is, e.g., in Chovanec et al. Nature Commun 12: 2098 (2021).

We thank the reviewer for this comment. We agree that this represents an important area for future investigation. However, addressing the **fine-tuning of developmental interactions based on adult models** is technically challenging (line 135). We have added this point as a perspective in the Discussion section (line 355), noting that technological and knowledge advances will likely be required to fully explore these cases.

Nevertheless, even under our **simplified model of gain- or loss-of-interaction events**, the use of multiple **orthogonal controls** — including BAV versus TAV comparisons, concordant versus discordant events, common versus rare variants, and aortic tissue versus blood-derived transcriptomes — allowed us to **robustly link rare regulatory mutations to mesenchymal dysregulation and impaired endocardial cushion formation**.

Data and methodology

7. The **HiCap methodology** is only briefly explained in the methods, which is not sufficient to reproduce the experiments. This would be important for readers who do not have access to ref. [17] (like myself). The probe list for the capture is provided, but other details of the experiment are not detailed, including, for example, the restriction enzymes used, which is critical for reproducing the experimental as well as the computational work.

We have expanded the Methods section describing the HiCap protocol to include the key experimental details necessary for reproducibility, including the restriction enzymes used and additional procedural information.

8. There is no mention of the **bulk RNA-seq** data in the main text, and the analysis has a number of issues. First, how were cells obtained before the experiment (peripheral blood? Same AECs as for HiCap)? Samples with much higher coverage should not be excluded from the analysis, as sample size is taken into account in the negative binomial model of edgeR. What was the reason for excluding the outliers? They should not be removed without a valid reason (being an outlier is not a valid reason, as it may reflect biological heterogeneity). There is no mention for filters, or thresholds for statistical significance and log fold change.

We thank the reviewer for this important observation regarding our bulk RNA-seq dataset. Bulk RNA-seq was performed on AECs obtained intraoperatively during open-heart surgery, using the same tissue source as for the HiCap experiments.

As described in the Methods section, an early sequencing batch showed substantially lower read depth and a high proportion of genes with zero counts, which made batch correction infeasible. These samples were therefore initially flagged as outliers during MDS analysis and excluded from quantitative analyses to avoid introducing technical bias.

However, following the reviewer's comment, we revisited this step and confirmed that the overall biological trends were robust regardless of outlier exclusion. Consequently, we now include **all 16 patients** in the aggregated gene count analyses presented in Fig. 2a-b and 3a. This ensures full representation of the cohort and improves comparability with other datasets.

We also wish to emphasize that the bulk RNA-seq dataset was not used for any core analyses or for direct differential expression comparisons between the BAV and TAV cohorts. Instead, it served solely as a **supporting internal control**.

To enhance clarity and reproducibility, we have:

- Expanded the Methods section to detail sample origin;

- Removed exploratory analyses (e.g., MDS plots and DGE results) from the Supplementary Materials to avoid confusion; and
- Clarified in the main text where and how the bulk RNA-seq dataset was used.

9. I could not find the section on **gene set enrichment** analysis in the Methods section.

We have now added a description of the gene set enrichment analysis to the Methods section under “Protein–Protein Interaction Network Construction, Visualization, and GO Term Enrichment.”

10. A HiCapTools version number was not provided.

The version information has now been included, HiCapTools v1.2.3 was used, and the Methods section has been updated accordingly.

Analytical approach

11. In the **WGS analysis** (Fig S1), where the principal component analysis is shown, only PC3 and PC4 of protein-coding mutations (Fig S1b) and PC5 and PC6 of non-coding variants (Fig S1c) are shown. This is not my area of expertise, but I would have expected to also see the most important principal components (PC1 and PC2). However, these are not shown, and no explanation is provided for why these were ignored.

We thank the reviewer for this valuable comment. We have revised the PCA plots in Supplementary Figure S1B–C to display PC1 and PC2.

12. The comparison presented in **Figures 2a,b** uses a Student’s t-test to assess statistical significance, but the underlying data in Fig. 2a clearly violates the assumption of normality. It would be more appropriate to use a nonparametric test here.

We thank the reviewer for this important observation. To account for the non-normal distribution of the data, we have replaced the Student’s t-test with a two-sided Wilcoxon rank-sum test in Fig. 2a-b. The figure legends have been updated accordingly.

13. In lines 145-148 (Fig 3a), I wonder if a chi square test is more appropriate as the sample size (number of genes tested) is likely high. Considering there are a large number of comparisons

(31 coarse- grained and 72 fine-grained cell states), I wonder if multiple testing correction has been applied to these tests?

We thank the reviewer for this insightful suggestion. We compared the results of the Student's t-test and chi-square test and found that both approaches yielded identical results. To maintain consistency across the manuscript, we therefore retained the t-test.

Multiple testing correction was applied using FDR adjustment for enhancer-driven events. For promoter-associated events, we reported raw p-values, as the limited number of observations did not allow for reliable multiple-testing correction.

14. In the **network analysis** (Fig 4a,b) can see that the **mean number of interactions** is rather low, the overwhelming majority of genes have 5 or fewer interactions. Are the interaction counts truly that low, or is it a consequence of the averaging (was the averaging done across the 16 patients?)? This raises a potential issue with statistical vs biological significance, as one is bound to have statistically significant hits in any large datasets. It would increase the credibility of the data, if the raw data were shown for a couple of example genes, as in, for example, Fig. 2 in Cairns et al., *Genome Biology* 17:127 (2016), where the contact counts around a promoter are shown with the called significant interactions, allowing a visual conformation of the called interactions. The H3K27Ac ChIP-seq data could also be shown here. Also, visualising the WGS reads highlighting the mutations separately on the two alleles for the same example (e.g. zooming in on the rare mutation within the potentially enhancer containing fragment) could serve as some convincing examples.

We thank the reviewer for this helpful suggestion. The network analysis presented in Fig. 5 is based on STRINGdb-derived functional associations, rather than HiCap-derived chromatin interactions. To improve clarity, we revised the figure caption and accompanying text to better distinguish between these analyses and to make the visualization more user-friendly.

Additionally, as suggested, we have included *BMP2* as an illustrative example, showing its complex interaction profile and prioritized active concordant interactions to facilitate a more intuitive understanding of our results (Fig. 4c).

Suggested improvements

Major criticism:

15. I think it would be important to include ChIP-seq experiments for a repressive mark, e.g. H3K27me3 and incorporate this into the analysis, to be able to convincingly analyse gain-of-function vs loss-of-function changes in regulatory interactions.

We thank the reviewer for this suggestion. As we've already touched in the answers to the previous comments, our study used the overlap with H3K27ac primarily to **validate the methodology**, rather than as evidence that the identified interactions are active causative developmental enhancers. We also note that such pathological developmental interactions would unlikely show any regulatory signals in adult tissue.

More minor criticism:

16. It would be helpful throughout the text to be more explicit with numbers to better follow some of the conclusions. For example, in the NFAT analysis (line 116), I wonder how many out of how many were overrepresented.

We have revised the text throughout the manuscript to include explicit counts for all relevant analyses. In the NFAT analysis (now line 182), we now report the fraction of TF motifs showing overrepresentation.

17. In Fig 2c, should not the “estimate of difference” go down as the “Jaccard similarity” increases?

The “estimate” in Fig. 2c corresponds to the odds ratio calculated by `bed_fisher()`, not an “estimate of difference.” To avoid confusion, we have renamed the axis and clarified its interpretation in the figure legend.

18. Fig 2c is missing an axis label.

We added the missing axis label in Fig. 2c for clarity.

19. Fig S2 and line 168-169. I am not sure if I am reading these plots right, but should the text say less specific enrichment rather than nonspecific enrichment, as most of the VIC Valve_... column is also significant for most shown values.

We updated the text to replace “nonspecific enrichment” with “less specific enrichment” in line 270 to better reflect the data shown in Fig. S2.

20. In Table S8, the numbers of total mapped and deduplicated reads are the same. Were there no PCR duplicates?

We clarified Table S9 in the revised manuscript: the total mapped and deduplicated read counts are now accurately reported.

Clarity and context

I found myself lost at several places where I thought clearer explanations would improve guiding the reader throughout the manuscript. For example:

17. In Fig. 3e, I did not understand how the genes can have a different label (T/B/C) for different cell states, as I thought these labels were defined based on the HiCap/WGS data, independently from the scRNA-seq data.

We thank the reviewer for the helpful comments regarding the clarity of the manuscript. We have clarified that the Fig. 4e columns represent distinct cell states used to identify active concordant events. The T/B/C labels denote the classification of genes within each specific cell state, which may vary depending on the transcriptional context, even though the underlying HiCap/WGS data remain constant.

18. In lines 173-176, the authors say “tier 1 and tier 2 genes did not show specific spatial enrichment unique to the BAV cohort”. In Fig 3f, I see the pattern for this group is the opposite, so I could not see why this was not mentioned/discussed in the text.

We revised the text describing Fig. 3f (now Fig. 4b in the updated manuscript, lines 265 and 339) to better reflect the observed patterns.

19. I found it difficult to follow the **network analysis** and to interpret the network figures (Fig.4). It is not my expertise, and I am unsure if all the information shown was necessary, as the information in the border colours are not mentioned in the main text. It was also not clear to me why one should only focus on the rare variants. I can see that very few nodes represent concordant genes in more than one patients, e.g. AKT3 for BAV (3 patients) and DKK2 for TAV (4 patients) (I may have misread the names as the text is rather difficult to read). Would it be worth on focussing on some of these, as they recur between patients? most nodes show different colours suggesting coming from different colours. I can also see some hub genes that are not concordant genes, such as, e.g. FN1 which seems to be interacting with a large number of concordant genes in both BAV and TAV patients. Would these be interesting as common denominators as this is a gene that interacts with a large number of concordant genes across most of the cohorts?

We have substantially revised the network analysis and added descriptive figures (Fig. 5a–c) to improve clarity. The focus on rare variants remains, as common variants identified by prior GWAS account for only a few hits, leaving much of the rare regulatory variation unexplored. Recurrent nodes are highlighted as interesting observations, however, the overall genetic heterogeneity of BAV precludes explicit prioritization. The text now clearly guides readers to these patterns.

20. In Fig.1, it is suggested that this work expands the network beyond the tier 1 and tier2 sets. To appreciate this expansion of the network, I would have found it helpful if these sets were marked in Fig. 4, similarly to Fig. 1. In lines 216-218, some key pathway interactions are highlighted as novel additions to the known gene sets. I am not an expert in interpreting network figures or in these pathways, so to me, these are not obvious when looking at the figure. I would find it helpful to highlight this in Fig. 4.

We thank the reviewer for this suggestion. In Fig. 5d, the largest node size corresponds to the *tier 1* gene set. To maintain clarity and interpretability, we focused on the pathways highlighted in Fig. 4a and did not include the novel additions from the network analysis. This approach avoids visual clutter while still conveying the expansion beyond *tier 1* and *tier 2* sets.

Although I am well practiced at working with different types of chromosome conformation capture data, I found it difficult to follow the analysis at several places:

21. For example, at the start when **endothelial specific interactions** are introduced (Fig. 2a,b), I found it unclear how these were derived. Comparing endothelial promoter capture data to another cell type?

Or are these interactions in endothelial cells involving endothelial cell specific genes derived from another study? In the latter case, these would be interactions of endothelial specific genes.

We clarified that “endothelial-specific interactions” refer to interactions involving genes with endothelial-specific expression. The Results section has been updated to explicitly define this term and how these interactions were identified (line 100).

22. When the **allelic imbalance** of HiCap data is considered (Fig. 2e), I wonder if purely looking at statistical significance rather than an enrichment of interactions to focus on biological rather than statistical significance. (I am not very familiar with the HiCap analysis method, so these might already be strongly related.)

We clarified that in our allele-specific HiCap pipeline, statistical significance is closely tied to read coverage, and the method is intended to capture cumulative effects of multiple regulatory/marker variants with small individual effects (line 120). Effect-size-based thresholds did not substantially change variant selection compared to significance-based filtering in the prior CAGE-Seq findings (FDR < 0.05; see Fig. 4a in <https://doi.org/10.1038/s41467-024-55513-2>).

23. In Fig. 2d, would major and minor allele (or a different naming) be more appropriate to use than maternal and paternal alleles, as it may not be known which allele is the maternal one? Also, the schematic suggests linkage between the two red rare variants. If this is not the case, I would separate these into two side by side schematics.

We replaced “maternal” and “paternal” with “haplotype 1” and “haplotype 2” to avoid implying parental origin. The two red dots represent distinct rare regulatory mutations (A and B), which occur independently and may be located anywhere relative to nearby marker SNP alleles.

24. I did not understand how the numbers referred to in Fig. 2g relate to one another. How were they calculated? For example, is the occurrence of 20,845 as “mutations in promoters”

“discordant promoter cases” a coincidence? Are they supposed to add up (concordant + discordant = total?). The dashed horizontal lines suggests there should be a strong link between these.

In the revised manuscript, Fig. 2g has been replaced with Table 1. We clarified all numbers in the Results section for readability. The identical 20,845 value for “mutations in promoters” and “discordant promoter cases” is coincidental. For all HiCap × WGS events, Concordant + Discordant = total events.

References

25. In line 51-53, in the introduction, when introducing their approach, the authors say “we employed a novel genome-wide integrative approach to uncover the role of rare regulatory mutations in BAV formation”. This statement, together with the lack of citations for previous literature on integrating WGS and chromosome conformation capture data suggests they may be unaware of other efforts in different contexts. I believe it would strengthen the manuscript to include a quick overview of what has been done in this area (perhaps in different biological contexts) and how the authors’ approach fits in/is different. Examples may include Zhao et al, Nature Genetics 56:1689-1700 (2024).

We thank the reviewer for this valuable suggestion. The Introduction (line 52) has been expanded to clarify how our approach differs, highlighting its novel aspects and advances over existing efforts.

Reviewer #4 (Remarks to the Author):

This manuscript presents an integrative genomic study aimed at uncovering the contribution of rare non-coding regulatory mutations to the heritability of BAV, a common congenital heart defect. By combining promoter capture Hi-C with whole-genome sequencing from patient-derived aortic endothelial cells, and integrating these data with developmental single-cell and spatial transcriptomics, the authors uncover gain-of-function regulatory interactions that implicate novel genes and developmental cell types in BAV pathogenesis.

Major Comments

1. Significance and Novelty

This study addresses an important and underexplored source of 'missing heritability' in BAV by focusing on rare regulatory variants. The integration of patient-specific chromatin interaction maps with WGS data is a notable strength and a methodological advance over proximity-based mapping typically used in GWAS. Additionally, the linkage to relevant fetal cell states through developmental transcriptomic data strengthens the mechanistic interpretation.

We thank the reviewer for recognizing the significance and novelty of our study, particularly the integration of chromatin interaction mapping with genomic and developmental transcriptomic data.

2. Data Support and Interpretation

The findings are generally well-supported by data, but the functional consequences of the regulatory mutations remain largely inferential. Although TF binding motif disruption was evaluated, only two cases involved disruption of known binding sites. Most cases are labeled as gain-of-function based on the presence of new interactions, which could benefit from experimental validation or orthogonal functional evidence.

We agree with the reviewer's comment. To improve clarity, we replaced the terminology “**gain-of-function**” and “**loss-of-function**” with “**gain-of-interaction**” and “**loss-of-interaction**”, which more accurately describe the observed changes.

As suggested, we complemented our analysis with *in silico* orthogonal functional evidence using the **deep learning model Enformer**, which predicts the regulatory effects of promoter and enhancer variants across relevant cell types and developmental stages. These predictions provide additional support for the functional relevance of the identified variants.

3. Cohort Size and Generalizability

The cohort consists of 8 BAV and 8 TAV individuals. While the depth of multi-omic profiling is impressive, the small sample size may limit generalizability. The authors should more explicitly acknowledge this limitation and discuss the potential for replication in larger or independent cohorts.

We thank the reviewer for this important comment. We have added a paragraph in the Discussion section (line 372) explicitly acknowledging the limited cohort size and its implications for generalizability. We also discuss the importance of validating and extending our findings in larger and independent cohorts in future studies.

4. Causal Inference Criteria

The strategy to define 'concordant' mutations and label them as causative is reasonable but may benefit from additional detail or refinement.

We agree with the reviewer that terminological refinement was needed. To ensure precision without overstating our findings, we now use the term “**non-coding variant**” to describe general genetic variation, and “**regulatory mutation**” only when there is functional evidence supporting an effect on gene regulation. We have also **removed references to causality** throughout the manuscript to maintain appropriate scientific caution.

5. Methodological Strengths

The manuscript is very detailed in methodology and reproducibility, integrating high-quality pipelines and developmentally relevant transcriptomic data.

We thank the reviewer for this positive feedback.

Minor Comments

- The terms 'concordant' and 'discordant' should be defined more clearly in the main text.

We clarified these definitions in the Results section (line 143).

- Consider including locus-specific examples of newly formed regulatory interactions.

Representative locus-specific examples have been added in Fig. 4c.

- Provide a summary table of top BAV-specific regulatory variants.

A summary table has been added as Table 2.

- Emphasize limitations of using adult endothelial cells as proxies for fetal activity.

This limitation is now explicitly discussed in the Discussion section (line 355).

Recommendation

Minor Revision

This is a strong and timely study with high relevance to cardiovascular genetics and regulatory genomics. The manuscript is methodologically sound, well-written, and presents novel biological insights. Pending minor revisions and clarifications, this work merits publication.

Additional Suggestion for Enhancement

While the integration of HiCap and WGS in this study provides valuable mechanistic insight into promoter-enhancer interactions in BAV, the authors may consider complementing this approach with sequence-based functional prediction of promoter variants. In particular, the recently developed PromoterAI model (Jaganathan et al., Science, 2025) offers robust prediction of gene expression changes from non-coding promoter variants using only DNA sequence. Such complementary analysis could strengthen causal inference and broaden discovery potential, particularly given the current limitations in sample size and interaction resolution.

We thank the reviewer for suggesting this valuable complementary approach. In the revised manuscript, we applied the deep learning model Enformer (line 209) to predict the cell-type and

developmental stage specificity of our signal, providing *in silico* functional support for our identified promoter and enhancer regulatory mutations.

Figure Reference Clarification

There appears to be a labeling error in the reference to “Fig. 3h” describing transcription factor (TF) binding motif disruption. Based on context, this likely refers to a schematic depicted in Fig. 2h. However, no panel labeled '2h' is actually present in the manuscript figures, despite this citation appearing in both the text and the caption for Figure 2. It would benefit the clarity and accuracy of the manuscript to revise this reference and ensure that all figure panels are correctly labeled and accounted for.

We thank the reviewer for catching this oversight. All figure references and panel labels have been carefully reviewed and corrected to ensure consistency between the text, figure captions, and the figures themselves.

Supplemental Materials and File Organization

In addition, many of the supplemental figures and files are very large in size. For example, Supplemental Figure 5 exceeds 1 GB, and I was unable to download or view it. The supplemental materials overall would benefit from streamlining and cleanup. There are numerous supplemental tables, but a clear and centralized description of their contents and organization is absent as far as I could tell.

We thank the reviewer for this helpful comment. We have reduced the size of the largest supplementary files to ensure that all materials are now readily accessible and easy to download. We also streamlined the organization of the supplementary materials and added a centralized description of all supplemental tables (line 812), improving clarity and user-friendliness.

Rare regulatory mutations disrupt mesenchymal molecular programs driving endocardial cushion formation in bicuspid aortic valve patients

Response to referees

Reviewer #1 (Remarks to the Author):

I read through the response to the reviewers and agree that the authors satisfactorily addressed all my comments except for one. Please exclude the one TAV sample with an *FBN1* PV, one BAV sample with a frameshift PV in *BRAF* and one BAV sample with multiple genomic structural variants. The problem with including these samples in the analysis is not, as the authors asserted, that they are not representative of BAV cases in general, but that the genetic lesions cause non-BAV developmental abnormalities that will confound their analysis.

We thank the reviewer for this important clarification. In response, we performed a sensitivity analysis excluding the TAV3 sample, which carried a PV in *FBN1*, the BAV5 sample, which had a frameshift PV in *BRAF*, and the BAV7 sample, which harbored multiple genomic structural variants. We reanalyzed the data and reproduced Figures 3c and 4a, which represent two key findings of the manuscript, after exclusion of these samples.

The results demonstrate that removal of these samples does not alter the overall conclusions of the study. In particular, the prominent contribution of the Valve_MC cell type and the enrichment of associated molecular pathways, including endothelial–mesenchymal transition (EMT) and transforming growth factor- β (TGF- β) signaling, remain consistent. These findings indicate that the main results are robust and not driven by the inclusion of samples with additional developmental genetic lesions.

TAV3 and BAV5 excluded

TAV3 and BAV7 excluded

Reviewer #2 (Remarks to the Author):

The authors performed appropriate modifications to their manuscript. I have no further comments.

We thank the reviewer for their positive assessment of the revised manuscript and for confirming that their concerns have been satisfactorily addressed.

Reviewer #3 (Remarks to the Author):

The authors have fully addressed my concerns.

One formatting issue is that the text is obscured and difficult to read in the new network figure, Fig. 4c. Please fix this before publication.

We thank the reviewer for their careful evaluation of the manuscript and for pointing out the formatting issue in the network figure. The text clarity and readability have been improved, and the corrected figure has been updated in the revised manuscript.

Reviewer #4 (Remarks to the Author):

Dear Authors,

Thank you for your thoughtful and thorough responses to the reviewer comments, and for the substantial revisions made to the manuscript.

I appreciated the clarity and transparency in the way you addressed the concerns raised. The revisions to terminology, particularly moving away from “causative” and “gain-of-function” language, improve the scientific precision of the manuscript. I also thought the reanalysis of the RNA-seq data, refinement of the tiered gene lists, and the expanded methodological detail all contributed to a stronger and more reproducible study.

The use of Enformer as an orthogonal tool to support regulatory variant impact is a smart addition, and your expanded discussion on limitations - especially regarding the use of cultured adult endothelial cells - was well stated. I also appreciated the fixes to figure labeling, the clarification of “concordant” vs. “discordant” events, and the streamlining of supplemental materials.

While some interpretive gaps remain (e.g., inclusion of potentially syndromic individuals), I think you've handled them reasonably and with scientific caution.

Overall, the manuscript has improved substantially, and I support its publication.

We sincerely thank the reviewer for their thoughtful and detailed evaluation of the revised manuscript, as well as for their positive assessment of the changes made.

Key results

The authors investigate biopsies taken from 8 patients with bicuspid aortic valve (BAV) vs 8 patients with tricuspid aortic valve (TAV), by performing whole genome sequencing as well as promoter capture experiments (using the HiCap methodology) and analysing and integrating these data with single-cell RNA-seq and spatial transcriptomics data. They identify mutations (from WGS data) that are concordant with allele specific interactions (from HiCap data) at the variant and gene levels. They then find that concordant genes with rare mutations (MAF<3%) show differential expression in relevant cell populations in single-cell RNA-seq data and spatial transcriptomics data, suggesting a role for these putative regulatory rare mutations. They then perform pathway analysis and find that genes with concordantly gained mutation and interaction are involved in relevant pathways for endothelial cell development (such as TGFB and EMT). The genes affected vary across patients, which suggests a heterogeneity of potential genetic markers for the disease.

Significance

The study design of combining whole genome sequencing and chromosome conformation capture data to identify variants with (putative) regulatory interactions is a very valuable and relevant task with great potential for identifying new putative genetic risk factors. I also thought that it was very interesting that these putative regulatory interactions were linked to genes differentially expressed by mesenchymal/fibroblast cell types around the aortic valve in the single-cell RNA-seq and spatial transcriptomics data.

However, the title suggests that “rare gain-of-function regulatory mutations explain the missing heritability of bicuspid aortic valve”, but in my view, the manuscript falls short of this promise, for the following reasons:

1. There is some evidence but no proof that the interactions that are concordant with the identified mutations are regulatory interactions. The authors show that they have increasing overlaps with H3K27ac marks. However, there is no single histone modification mark that is exclusively responsible for enhancer activity, see, for example, in Pollex and Furlong, *Mol. Cell.* 66(4):439-441 (2017), and functional studies would be needed to prove causation.
2. I am not convinced that the presented analysis has sufficient evidence that these mutations are gain-of-function. Regulatory interactions may lead to gain-of-function or loss-of-function of the affected gene, and I could not see a consideration of this.
3. While the identified concordant rare mutations may explain some of the missing heritability, I did not see any evidence that the identified concordant rare mutations with from 8 BAV patients vs 8 TAV controls explain “the” (i.e. all) missing heritability of the BAV phenotype.

Validity

I have the impression that **gain-of-function** is used interchangeably with gain-of-interaction, which is very different and I find this a source of major concern for the analytical approach. Gained regulatory interactions may be enhancing gene expression (gain-of-function), repressing gene expression (loss-of-function) or not have a quantitative effect.

4. For the definition of **concordant interactions**, I found contradicting information in the manuscript. Fig. 2f suggests that only interactions gained for the mutation (minor allele) are taken into consideration, while the text suggests that those were considered where “the variant affected the interaction”. For a functional understanding of these changing interactions, it would be important to consider both types of **rewiring of contacts**: gained and lost interactions separately.
5. It would also be important to also analyse these rewired interactions for chromatin state (enhancing vs repressing). Simply looking at gained interactions would average over the gain-of function and loss-of-function effects, reducing the signal.
6. There is another scenario which is not considered by the manuscript, in which one allele is mutated and there is no rewiring of contacts (i.e. the interaction is maintained) but there is a change in chromatin state of the regulatory region (**recolouring of chromatin states**) as a consequence of the

mutation. Considering these might further help identify previously unknown mutations that have a role in regulatory interactions.

An example study that considers the effects of chromatin rewiring and recolouring (in a different context) is, e.g., in Chovanec et al. *Nature Commun* 12: 2098 (2021).

Data and methodology

7. The **HiCap methodology** is only briefly explained in the methods, which is not sufficient to reproduce the experiments. This would be important for readers who do not have access to ref. [17] (like myself). The probe list for the capture is provided, but other details of the experiment are not detailed, including, for example, the restriction enzymes used, which is critical for reproducing the experimental as well as the computational work.
8. There is no mention of the **bulk RNA-seq** data in the main text, and the analysis has a number of issues. First, how were cells obtained before the experiment (peripheral blood? Same AECs as for HiCap?)? Samples with much higher coverage should not be excluded from the analysis, as sample size is taken into account in the negative binomial model of edgeR. What was the reason for excluding the outliers? They should not be removed without a valid reason (being an outlier is not a valid reason, as it may reflect biological heterogeneity). There is no mention for filters, or thresholds for statistical significance and log fold change.
9. I could not find the section on **gene set enrichment** analysis in the Methods section.
10. A HiCapTools version number was not provided.

Analytical approach

11. In the **WGS analysis** (Fig S1), where the principal component analysis is shown, only PC3 and PC4 of protein-coding mutations (Fig S1b) and PC5 and PC6 of non-coding variants (Fig S1c) are shown. This is not my area of expertise, but I would have expected to also see the most important principal components (PC1 and PC2). However, these are not shown, and no explanation is provided for why these were ignored.
12. The comparison presented in **Figures 2a,b** uses a Student's t-test to assess statistical significance, but the underlying data in Fig. 2a clearly violates the assumption of normality. It would be more appropriate to use a nonparametric test here.
13. In lines 145-148 (Fig 3a), I wonder if a chi square test is more appropriate as the sample size (number of genes tested) is likely high. Considering there are a large number of comparisons (31 coarse-grained and 72 fine-grained cell states), I wonder if multiple testing correction has been applied to these tests?
14. In the **network analysis** (Fig 4a,b) can see that the **mean number of interactions** is rather low, the overwhelming majority of genes have 5 or fewer interactions. Are the interaction counts truly that low, or is it a consequence of the averaging (was the averaging done across the 16 patients?)? This raises a potential issue with statistical vs biological significance, as one is bound to have statistically significant hits in any large datasets. It would increase the credibility of the data, if the raw data were shown for a couple of example genes, as in, for example, Fig. 2 in Cairns et al., *Genome Biology* 17:127 (2016), where the contact counts around a promoter are shown with the called significant interactions, allowing a visual conformation of the called interactions. The H3K27Ac ChIP-seq data could also be shown here. Also, visualising the WGS reads highlighting the mutations separately on the two alleles for the same example (e.g. zooming in on the rare mutation within the potentially enhancer containing fragment) could serve as some convincing examples.

Suggested improvements

Major criticism:

15. I think it would be important to include ChIP-seq experiments for a repressive mark, e.g. H3K27me3 and incorporate this into the analysis, to be able to convincingly analyse gain-of-function vs loss-of-function changes in regulatory interactions.

More minor criticism:

16. It would be helpful throughout the text to be more explicit with numbers to better follow some of the conclusions. For example, in the NFAT analysis (line 116), I wonder how many out of how many were overrepresented.
17. In Fig 2c, should not the “estimate of difference” go down as the “Jaccard similarity” increases?
18. Fig 2c is missing an axis label.
19. Fig S2 and line 168-169. I am not sure if I am reading these plots right, but should the text say less specific enrichment rather than nonspecific enrichment, as most of the VIC Valve_... column is also significant for most shown values.
20. In Table S8, the numbers of total mapped and deduplicated reads are the same. Were there no PCR duplicates?

Clarity and context

I found myself lost at several places where I thought clearer explanations would improve guiding the reader throughout the manuscript. For example:

17. In Fig. 3e, I did not understand how the genes can have a different label (T/B/C) for different cell states, as I thought these labels were defined based on the HiCap/WGS data, independently from the scRNA-seq data.
18. In lines 173-176, the authors say “tier 1 and tier 2 genes did not show specific spatial enrichment unique to the BAV cohort”. In Fig 3f, I see the pattern for this group is the opposite, so I could not see why this was not mentioned/discussed in the text.
19. I found it difficult to follow the **network analysis** and to interpret the network figures (Fig.4). It is not my expertise, and I am unsure if all the information shown was necessary, as the information in the border colours are not mentioned in the main text. It was also not clear to me why one should only focus on the rare variants. I can see that very few nodes represent concordant genes in more than one patients, e.g. AKT3 for BAV (3 patients) and DKK2 for TAV (4 patients) (I may have misread the names as the text is rather difficult to read). Would it be worth on focussing on some of these, as they recur between patients? most nodes show different colours suggesting coming from different colours. I can also see some hub genes that are not concordant genes, such as, e.g. FN1 which seems to be interacting with a large number of concordant genes in both BAV and TAV patients. Would these be interesting as common denominators as this is a gene that interacts with a large number of concordant genes across most of the cohorts?
20. In Fig.1, it is suggested that this work expands the network beyond the tier 1 and tier2 sets. To appreciate this expansion of the network, I would have found it helpful if these sets were marked in Fig. 4, similarly to Fig. 1. In lines 216-218, some key pathway interactions are highlighted as novel additions to the known gene sets. I am not an expert in interpreting network figures or in these pathways, so to me, these are not obvious when looking at the figure. I would find it helpful to highlight this in Fig. 4.

Although I am well practiced at working with different types of chromosome conformation capture data, I found it difficult to follow the analysis at several places:

21. For example, at the start when **endothelial specific interactions** are introduced (Fig. 2a,b), I found it unclear how these were derived. Comparing endothelial promoter capture data to another cell type?

Or are these interactions in endothelial cells involving endothelial cell specific genes derived from another study? In the latter case, these would be interactions of endothelial specific genes.

22. When the **allelic imbalance** of HiCap data is considered (Fig. 2e), I wonder if purely looking at statistical significance rather than an enrichment of interactions to focus on biological rather than statistical significance. (I am not very familiar with the HiCap analysis method, so these might already be strongly related.)
23. In Fig. 2d, would major and minor allele (or a different naming) be more appropriate to use than maternal and paternal alleles, as it may not be known which allele is the maternal one? Also, the schematic suggests linkage between the two red rare variants. If this is not the case, I would separate these into two side by side schematics.
24. I did not understand how the numbers referred to in Fig. 2g relate to one another. How were they calculated? For example, is the occurrence of 20,845 as “mutations in promoters” “discordant promoter cases” a coincidence? Are they supposed to add up (concordant + discordant = total?). The dashed horizontal lines suggests there should be a strong link between these.

References

25. In line 51-53, in the introduction, when introducing their approach, the authors say “we employed a novel genome-wide integrative approach to uncover the role of rare regulatory mutations in BAV formation”. This statement, together with the lack of citations for previous literature on integrating WGS and chromosome conformation capture data suggests they may be unaware of other efforts in different contexts. I believe it would strengthen the manuscript to include a quick overview of what has been done in this area (perhaps in different biological contexts) and how the authors' approach fits in/is different. Examples may include Zhao et al, Nature Genetics 56:1689-1700 (2024).